# Mitochondrial fusion and altered beta-oxidation drive muscle wasting in a *Drosophila* cachexia model

Callum Dark[1,2], Nashia Ali[1,10], Sofya Golenkina[1,10], Vaibhav Dhyani [ID] [3,4], Ronnie Blazev [ID] [5,6], Benjamin L Parker[5,6], Kate T Murphy[5,6], Gordon S Lynch [ID] [5,6], Tarosi Senapati[7], S Sean Millard [ID] [7], Sarah M Judge[8], Andrew R Judge [ID] [8], Lopamudra Giri[3], Sarah M Russell [ID] [2,4,9] & Louise Y Cheng[1,2,6✉]

## Abstract

Cancer cachexia is a tumour-induced wasting syndrome, characterised by extreme loss of skeletal muscle. Defective mitochondria can contribute to muscle wasting; however, the underlying mechanisms remain unclear. Using a *Drosophila* larval model of cancer cachexia, we observed enlarged and dysfunctional muscle mitochondria. Morphological changes were accompanied by upregulation of beta-oxidation proteins and depletion of muscle glycogen and lipid stores. Muscle lipid stores were also decreased in Colon-26 adenocarcinoma mouse muscle samples, and expression of the beta-oxidation gene *CPT1A* was negatively associated with muscle quality in cachectic patients. Mechanistically, mitochondrial defects result from reduced muscle insulin signalling, downstream of tumour-secreted insulin growth factor binding protein (IGFBP) homologue ImpL2. Strikingly, muscle-specific inhibition of Forkhead box O (FOXO), mitochondrial fusion, or beta-oxidation in tumour-bearing animals preserved muscle integrity. Finally, dietary supplementation with nicotinamide or lipids, improved muscle health in tumour-bearing animals. Overall, our work demonstrates that muscle FOXO, mitochondria dynamics/beta-oxidation and lipid utilisation are key regulators of muscle wasting in cancer cachexia.

**Keywords** *Drosophila*; Cachexia; Muscle; Lipid Metabolism
**Subject Categories** Cancer; Metabolism; Musculoskeletal System

## Introduction

Cancer cachexia is a metabolic wasting syndrome characterised by the involuntary loss of muscle and adipose tissue, caused by tumour-secreted factors, and is the cause of up to 20% of cancer-related deaths (Argilés et al, 2014). The most prominent clinical feature of cachexia is the continuous loss of skeletal muscle, which cannot be fully reversed by conventional nutritional support.

*Drosophila melanogaster* is emerging as an excellent model to identify tumour-secreted factors that drive cancer cachexia. Adult and larval tumour models that induce cachectic phenotypes have been established in *Drosophila*, and these models have revealed several mechanisms by which tumours induce these phenotypes (Kwon et al, 2015; Song et al, 2019; Lodge et al, 2021; Newton et al, 2020; Santabárbara-Ruiz and Léopold, 2021; Hodgson et al, 2021; Figueroa-Clarevega and Bilder, 2015; Ding et al, 2021; Khezri et al, 2021; Lee et al, 2021). In mouse and *Drosophila* models of cachexia, it has been shown that tumours can induce a number of metabolic alterations in the muscle (Bonetto et al, 2016; Judge et al, 2014), such as increased autophagy (Lodge et al, 2021), proteolysis (Newton et al, 2020), decreased protein synthesis (Lodge et al, 2021; Newton et al, 2020), defective mitochondrial function (Kwon et al, 2015; Figueroa-Clarevega and Bilder, 2015), reduced ATP levels (Kwon et al, 2015; Figueroa-Clarevega and Bilder, 2015) and depleted Extra Cellular Matrix (ECM) (Lodge et al, 2021; Bakopoulos et al, 2023). While all these changes are symptomatic of cachexia, it is not clear whether they directly drive cachexia; furthermore, it is unclear how these defects are linked to tumour-secreted factors.

In this study, utilising our previously characterised eye imaginal disc tumour models (Lodge et al, 2021), we identified two muscle-specific mechanisms that contribute to muscle wasting: (1) altered activity of the transcription factor Forkhead box O (FOXO), a negative regulator of insulin signalling and (2) increased beta-oxidation resulting from mitochondria fusion. Both mechanisms are mediated by a reduction of systemic insulin signalling induced by tumour-secreted insulin-like-peptide antagonist ImpL2 (Imaginal morphogenesis protein-Late 2). Downstream of FOXO, mitochondrial fusion (mediated by Mitochondrial assembly regulatory factor (Marf)) and increased beta-oxidation (mediated by Withered (Whd)) were responsible for increased utilisation of muscle lipids. Strikingly, inhibiting FOXO, Marf or Whd

[1]Peter MacCallum Cancer Centre, Melbourne, VIC 3000, Australia. [2]Sir Peter MacCallum Department of Oncology, The University of Melbourne, Melbourne, VIC 3010, Australia. [3]Bioimaging and Data Analysis Lab, Department of Chemical Engineering, Indian Institute of Technology Hyderabad, Sangareddy, Telangana, India. [4]Optical Science Centre, Faculty of Science, Engineering & Technology, Swinburne University of Technology, Hawthorn, Melbourne, VIC, Australia. [5]Centre for Muscle Research, The University of Melbourne, Melbourne, VIC 3010, Australia. [6]Department of Anatomy and Physiology, The University of Melbourne, Melbourne, VIC 3010, Australia. [7]School of Biomedical Sciences, Faculty of Medicine, The University of Queensland, Queensland, QLD 4072, Australia. [8]Department of Physical Therapy, College of Public Health and Health Professions, University of Florida, Florida, FL 32603, USA. [9]Immune Signalling Laboratory, Peter MacCallum Cancer Centre, Melbourne, VIC 3000, Australia. [10]These authors contributed equally: Nashia Ali, Sofya Golenkina. ✉E-mail: louise.cheng@petermac.org

specifically in the muscle was sufficient to over-ride the effects of tumour-secreted factors and improve muscle morphology when tumours were present. In addition, feeding cachectic animals a diet supplemented with nicotinamide (Vitamin B3), or a high-fat coconut oil diet, which replenished lipids in the muscle, were sufficient to improve muscle integrity. Finally, we show that these processes are important in other cachexia models. We observed a similar depletion of lipid reserves in a C-26 mouse cachexia model; furthermore, the beta-oxidation gene Carnitine Palmitoyltransferase 1A (*CPT1A*, the mammalian homologue of Whd) is negatively correlated with muscle quality in cachectic patients with pancreatic ductal adenocarcinoma. Together, our study shows that mitochondrial fusion, beta-oxidation and lipid utilisation are likely early hallmarks of muscle disruption during cancer cachexia.

# Results

## Inhibition of mitochondrial fusion prevents muscle detachment in tumour-bearing animals

Studies examining the muscles of cachectic patients have demonstrated increased mitochondrial energy expenditure (Dolly et al, 2022) and size (Dolly et al, 2022; de Castro et al, 2019), however, how changes in mitochondrial morphology underly cachexia is so far not clear. In this study, we utilise two *Drosophila* larval tumour models to study muscle biology. In the first model, the tumour is induced via the GAL4-UAS mediated overexpression of *Ras^V12^* and Disc Large (Dlg1) RNAi in the eye (Fig. 1A). In the second model, the tumour is induced via the QF2-QUAS mediated overexpression of *Ras^V12^* and *scrib* RNAi (Fig. 1A) in the eye, allowing us to knockdown or overexpress genes of interest in the muscles of tumour-bearing animals using drivers such as *MhcGAL4* or *Mef2GAL4* (Fig. 1A). Using these models, we have previously shown that tumours caused a loss of muscle integrity (Lodge et al, 2021).

Using Electron Microscopy (EM), we observed fewer and larger mitochondria in the sub-sarcolemma of muscles of *Ras^V12^dlg1^RNAi^* tumour-bearing animals at 7 days after egg lay (AEL, Fig. 1B,C). A disruption of muscle mitochondria morphology in the sub-sarcolemma plane of the muscle (as depicted in a cartoon of a muscle section, Fig. 1D) was further confirmed by antibody staining against mitochondria protein ATP5A, where we observed a gradual increase in mitochondrial size from 5 days AEL in both *Ras^V12^dlg1^RNAi^* and Q*Ras^V12^;scrib*^RNAi^ tumour models (Figs. 1E–I and EV1A–C), measured using a segmentation tool (for details on mitochondrial size quantification, see "Methods"). To test whether this increase in mitochondrial size is correlated with altered mitochondrial membrane potential, we performed live staining with tetramethylrhodamine ethyl ester (TMRE), a compound used to measure the mitochondrial membrane potential (Scaduto and Grotyohann, 1999). We detected a significant reduction in TMRE fluorescence in the muscles of tumour-bearing animals, indicative of reduced membrane potential (*Ras^V12^dlg1^RNAi^*, Figs. 1J–L and EV1D,E", Q*Ras^V12^scrib^RNAi^*, Fig. EV1F–H). Mitochondria perform oxidative phosphorylation via OXPHOS and is the main site for cellular energy conversion. Consistent with the reduction in membrane potential of the mitochondria, we detected a reduction in levels in the muscles of *Ras^V12^dlg1^RNAi^* and Q*Ras^V12^scrib^RNAi^*

tumour-bearing animals (Figs. 1M and EV1I). It is known that tumour-bearing animals exhibit a prolonged period of larval development (Caldwell et al, 2005). However, the differences we observed in muscle mitochondrial morphology and function, are not accounted for by developmental delay. No significant changes in mitochondria were detected in animals where we inhibited the prothoracicotropic hormone receptor *torso* in the prothoracic gland (*PhmGAL4*) that resulted in extended third instar (Rewitz et al, 2009) (Fig. EV1J–L).

Enlarged mitochondria has been shown to be attributed to mitochondrial fusion, a process evoked to buffer against mitochondrial damage caused by increased reactive oxygen species (Shutt et al, 2012; Godenschwege et al, 2009). We found that there was a significant increase in ROS levels (as indicated by dihydroethidium (DHE) staining) in the muscles of tumour-bearing animals (Q*Ras^V12^scrib^RNAi^*, Fig. EV1M–O). To determine if reducing ROS could preserve muscle morphology, we over-expressed ROS scavengers Catalase (Cat) and superoxide dismutase (Sod1), or antioxidative enzyme glutathione peroxidase 1 (GPx1, previously validated in (Owusu-Ansah and Banerjee, 2009; Ohsawa et al, 2012; Lim et al, 2014)), specifically in the muscles of tumour-bearing animals (Q*Ras^V12^scrib^RNAi^*). However, these manipulations did not rescue muscle integrity (Fig. EV1P–U), suggesting that ROS was a consequence but not the cause of muscle wasting.

Next, we asked if preventing mitochondrial fusion through muscle-specific knockdown of Marf, a protein important for outer mitochondrial membrane fusion(Sandoval et al, 2014a), could improve muscle integrity in tumour-bearing animals (Q*Ras^V12^scrib^RNAi^*). Marf knockdown in the muscle did not affect the tumour size (Fig. EV2A) but was effective in improving muscle integrity (Fig. 1N–Q), reducing mitochondrial size (Figs. 1R–T and EV2B) and increasing mitochondrial membrane potential (TMRE assay, Fig. 1U). However, Marf inhibition did not significantly affect ATP levels (Fig. EV2C). This is perhaps not surprising, as it has been previously reported that the muscles of *Drosophila* Marf mutants have smaller mitochondria and thus perform less energy conversion (Sandoval et al, 2014b). On the other hand, *marf* overexpression significantly worsened muscle detachment in tumour-bearing animals (Fig. EV2D–F). In addition, knockdown of Optic atrophy 1 (Opa1), a protein involved with inner mitochondrial membrane fusion, was also able to improve muscle integrity in tumour-bearing animals (Q*Ras^V12^scrib^RNAi^*, Fig. 1V–X) (van der Bliek et al, 2013). However, overexpression of mitochondrial fission protein Dynamin-related protein 1 (Drp1) (van der Bliek et al, 2013), did not help preserve muscle integrity (Q*Ras^V12^scrib^RNAi^*, Fig. EV2G–I). Together, these experiments suggest that preventing mitochondrial fusion (rather than increasing fission) helps to improve muscle function.

## Reduced insulin signalling mediates changes in mitochondrial size

Mitochondria are known to change in size in response to environmental stressors, especially starvation (Gomes et al, 2011; Rambold et al, 2011). Upon subjecting wild-type larvae to 24 h of nutrient restriction (Starved), which is known to reduce systemic insulin signalling (Britton et al, 2002), we observed an increase in mitochondrial size (Figs. 2A–C and EV3A), that is reminiscent of

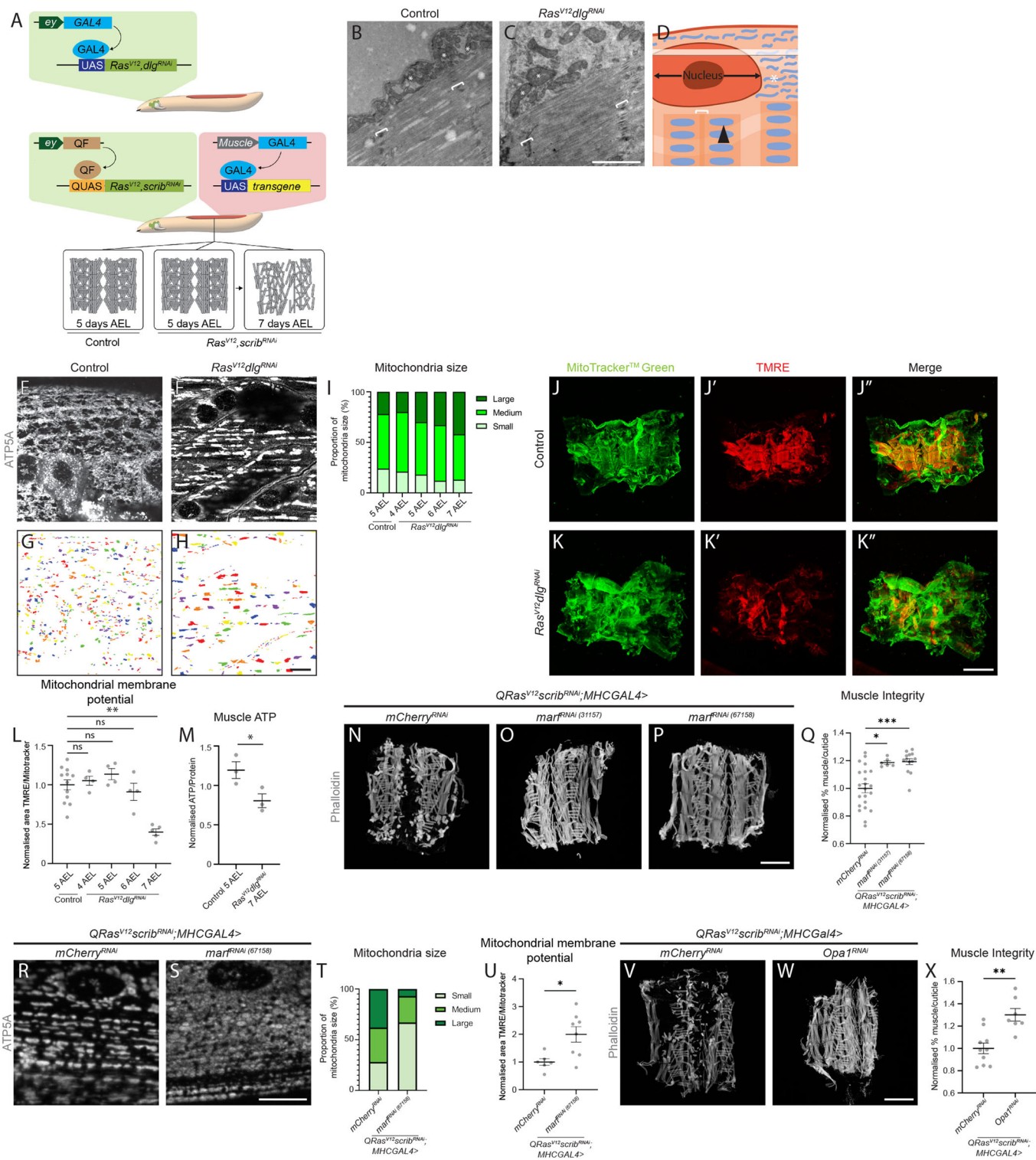

what we saw in the muscles of tumour-bearing animals. We previously reported that $Ras^{V12}dlg1^{RNAi}$ eye imaginal disc tumours express elevated levels of ImpL2, and its knockdown ameliorated muscle disruption without affecting tumour size (Lodge et al, 2021). Here, we found that tumour-specific inhibition of ImpL2 significantly reduced muscle mitochondrial size ($Ras^{V12}dlg1^{RNAi}$, Figs. 2D–F and EV3B), as well as increased muscle mitochondrial membrane potential (Fig. 2G), without affecting the muscle's ability to perform cellular energy conversion as reflected by ATP levels (Fig. EV3C). Together, these data suggest that mitochondrial fusion may be caused by reduced systemic insulin signalling.

**Figure 1. Blocking mitochondrial fusion in cachectic muscles is sufficient to restore muscle mitochondrial defects, as well as muscle health.**

(A) Cartoon schematic depicting the two tumour models used in this study. The first model utilises the GAL4-UAS system to overexpress $Ras^{V12}$ and $dlg1^{RNAi}$ in the eye discs of *Drosophila* larvae. The second model uses the QF2-QUAS system to overexpress $Ras^{V12}$ and $scrib^{RNAi}$ in the eye discs, which allows us to utilise the GAL4-UAS system to simultaneously drive genetic manipulations in the muscle. Both of these systems have comparable muscle integrity to controls at 5 days AEL, but show muscle detachment by 7 days AEL. (B, C) Electron microscopy images of mitochondria in the muscles of control (5 days AEL) and $Ras^{V12}dlg1^{RNAi}$ larvae (7 days AEL). White asterisks mark mitochondria in the sub-sarcolemma, white brackets indicate z-discs of the sarcomeres. (D) Cartoon representation of a cross-section through a larval muscle. The black arrow points to a mitochondrion found in the sarcomere, while white asterisks point to the mitochondria found in the sub-sarcolemma. The white bracket indicates the z-disc of the sarcomere. The transparent white rectangle indicates the plane at which the confocal images and mitochondrial quantifications were acquired in (at the level of the sub-sarcolemma). (E, F) Representative images of ATP5A staining of mitochondria in the muscles of control (5 days AEL) and $Ras^{V12}dlg1^{RNAi}$ larvae (7 days AEL). (G, H) Segmented representative images of mitochondria from (E, F). (I) Quantification of the proportion of small, medium, and large mitochondria as a percentage of total mitochondria in control at 5 AEL and $Ras^{V12}dlg1^{RNAi}$ larvae at days 4–7 AEL, performed using Chi-square test ($P = 0.017$, $n = 4, 4, 4, 4, 4$). (J–K'') Representative images of control (5 days AEL) and $Ras^{V12}dlg1^{RNAi}$ (7 days AEL) larval muscle fillets stained with MitoTracker™ Green which labels all mitochondria (J, K), and active mitochondria with TMRE (J'–K'). Zoomed in images of (J–K'') are in Fig. EV1D,E''. Merged images are shown in (J'', K''). (L) Quantification of the percentage of total mitochondria stained with MitoTracker™ Green that are also positive for TMRE in control (5 AEL, $n = 12$) and $Ras^{V12}dlg1^{RNAi}$ (4–7 AEL, $n = 4, 4, 4, 5$) performed using the Kruskal–Wallis test. (M) Quantification of normalised muscle ATP measured in control (5 AEL) and $Ras^{V12}dlg1^{RNAi}$ (7 AEL) larvae, performed using Student's *t* test ($n = 3, 3$). (N–P) Representative muscle fillets from $QRas^{V12}scrib^{RNAi};MhcGAL4>mCherry^{RNAi}$ (N), $QRas^{V12}scrib^{RNAi};MhcGAL4>marf^{RNAi\ (31157)}$ (O), and $QRas^{V12}scrib^{RNAi};MhcGAL4>marf^{RNAi\ (67158)}$ (P) larvae (all 7 days AEL), stained with Phalloidin to visualise actin. (Q) Quantification of muscle integrity in (N–P) performed using Brown–Forsythe ($n = 22, 6, 13$). (R, S) Representative images of ATP5A staining of mitochondria in the muscles of $QRas^{V12}scrib^{RNAi};MhcGAL4>mCherry^{RNAi}$ and $QRas^{V12}scrib^{RNAi};MhcGAL4>marf^{RNAi\ (67158)}$ larvae (all 7 days AEL). (T) Quantification of the proportion of small, medium, and large mitochondria as a percentage of total mitochondria in (R, S) performed using Chi-square test ($P < 0.0001$, $n = 3, 5$). (U) Quantification of the percentage of total mitochondria stained with MitoTracker™ Green that are shown to be active via TMRE incorporation in the muscles of 6 days AEL $QRas^{V12}scrib^{RNAi};MhcGAL4>mCherry^{RNAi}$ and $QRas^{V12}scrib^{RNAi};MhcGAL4>marf^{RNAi\ (67158)}$ larvae, performed using Student's *t* test ($n = 6, 8$). (V, W) Representative muscle fillets from $QRas^{V12}scrib^{RNAi};MhcGAL4>mCherry^{RNAi}$ and $QRas^{V12}scrib^{RNAi};MhcGAL4>Opa1^{RNAi}$ larvae (7 days AEL), stained with Phalloidin to visualise actin. (X) Quantification of muscle integrity in (V, W) performed using Student's *t* test ($n = 10, 7$). Scale bars: 1 μm for (B, C), 10 μm for (E–H, R, S), and 500 μm for (J, J', J'', K, K', K'', N, O, P, V, W). Data information: All error bars are $+/-$ SEM. *P* values are: ns (not significant), $P > 0.05$; \**P* < 0.05; \*\**P* < 0.01; \*\*\**P* < 0.001. Source data are available online for this figure.

FOXO transcription factors have been implicated in muscle atrophy in wild-type animals via ubiquitin-proteasome mediated mechanisms (Hunt et al, 2019) and have been shown to be activated in response to decreased insulin/IGF signalling. In cachectic muscles ($Ras^{V12}dlg1^{RNAi}$), we found a significant increase in nuclear FOXO signal, which first occurred at 5 days AEL (Fig. 2H–J). This preceded muscle disruption at 7 days AEL (Lodge et al, 2021), suggesting that FOXO may be a key upstream mediator of muscle wasting. Furthermore, tumour-specific ImpL2 inhibition was able to significantly reduce muscle FOXO levels at 5 days AEL ($Ras^{V12}dlg1^{RNAi}$, Fig. 2K).

As the overexpression of FOXO has been linked with increased expression of mitochondrial fusion proteins in mice (Cheng et al, 2009; Lidell et al, 2011; Pancrazi et al, 2015), we next assessed whether increased mitochondrial size in cachectic muscles was regulated by FOXO. Muscle-specific FOXO knockdown was able to reduce mitochondrial size in tumour-bearing animals ($QRas^{V12}scrib^{RNAi}$, Figs. 2L–N and EV3D). We observed significantly increased muscle integrity (Fig. 2O–R) and mitochondrial membrane potential (Fig. 2S), as well as improved muscle ATP levels (Fig. EV3E, $P = 0.08$), all without affecting tumour size (Fig. EV3F). Conversely, overexpression of dFOXO in the muscles of tumour-bearing animals ($QRas^{V12}scrib^{RNAi}$), caused precocious muscle detachment at 6 days AEL (Fig. 2T–W). This was not caused by a change in the attachment of muscles to the cuticle, as the intensity or localisation of Tiggrin at muscle attachment sites was unaltered (Fig. EV3G–J). However, it is likely that changes in insulin signalling causes muscle atrophy (Bakopoulos et al, 2023). Finally, the overexpression of dFOXO in wild-type muscles also caused a loss of muscle integrity (Fig. EV3K–M). Together, these results indicate that FOXO plays an important role in regulating muscle integrity via mitochondrial membrane potential downstream of tumour-induced signals.

## Disrupted autophagy and translation in cachectic muscles is not necessary for muscle degradation

Mitochondrial disruption has been reported to be accompanied by increased autophagy (Kroemer et al, 2010) and reduced protein translation(Samluk et al, 2019). Using a transgenic line that expresses a tandem autophagy reporter (UAS pGFP-mCherry-Atg8a) (Nezis et al, 2010) under the control of a muscle-specific driver (MhcGAL4), we found the ratio of mCherry vs. EGFP was significantly elevated in tumour-bearing animals ($QRas^{V12}scrib^{RNAi}$) at 6 days AEL (Fig. EV3N–P). This indicates there is an increased lysosomal degradation and autophagy in the muscles of tumour-bearing animals (Mauvezin et al, 2014) prior to the increase in mitochondrial size at day 7. Next, we examined the expression of the sarcomere structural protein Myosin Heavy Chain (Mhc), a previously reported proxy for protein synthesis (Kim and O'Connor, 2021). We found Mhc levels were significantly down-regulated at 6 days AEL in tumour-bearing animals ($Ras^{V12}dlg1^{RNAi}$, Fig. EV3Q–S). This was also confirmed by OPP protein synthesis assay ($Ras^{V12}dlg1^{RNAi}$, Fig. EV3T–V). We found that inhibition of tumour-secreted ImpL2 ($Ras^{V12}dlg1^{RNAi}$), was sufficient to reduce ATG8a levels by 7 days AEL (Fig. EV3W–Y), and Mhc levels by 5 days AEL (Fig. EV3Z–BB), confirming that muscle perturbations are driven by tumour-secreted ImpL2.

As disruptions to autophagy and protein synthesis preceded muscle degradation (at 7 days AEL (Lodge et al, 2021)), we functionally assessed whether inhibiting autophagy or enhancing translation could rescue muscle integrity ($QRas^{V12}scrib^{RNAi}$). However, the inhibition of autophagy via the expression of a RNAi against the protein kinase Atg1 (Fig. EV3CC–EE) which has previously been shown to reduce autophagy (Xu et al, 2015) was not able to prevent tumour-induced muscle degradation. Furthermore, the expression of a constitutively activated S6 kinase (S6K$^{CA}$),

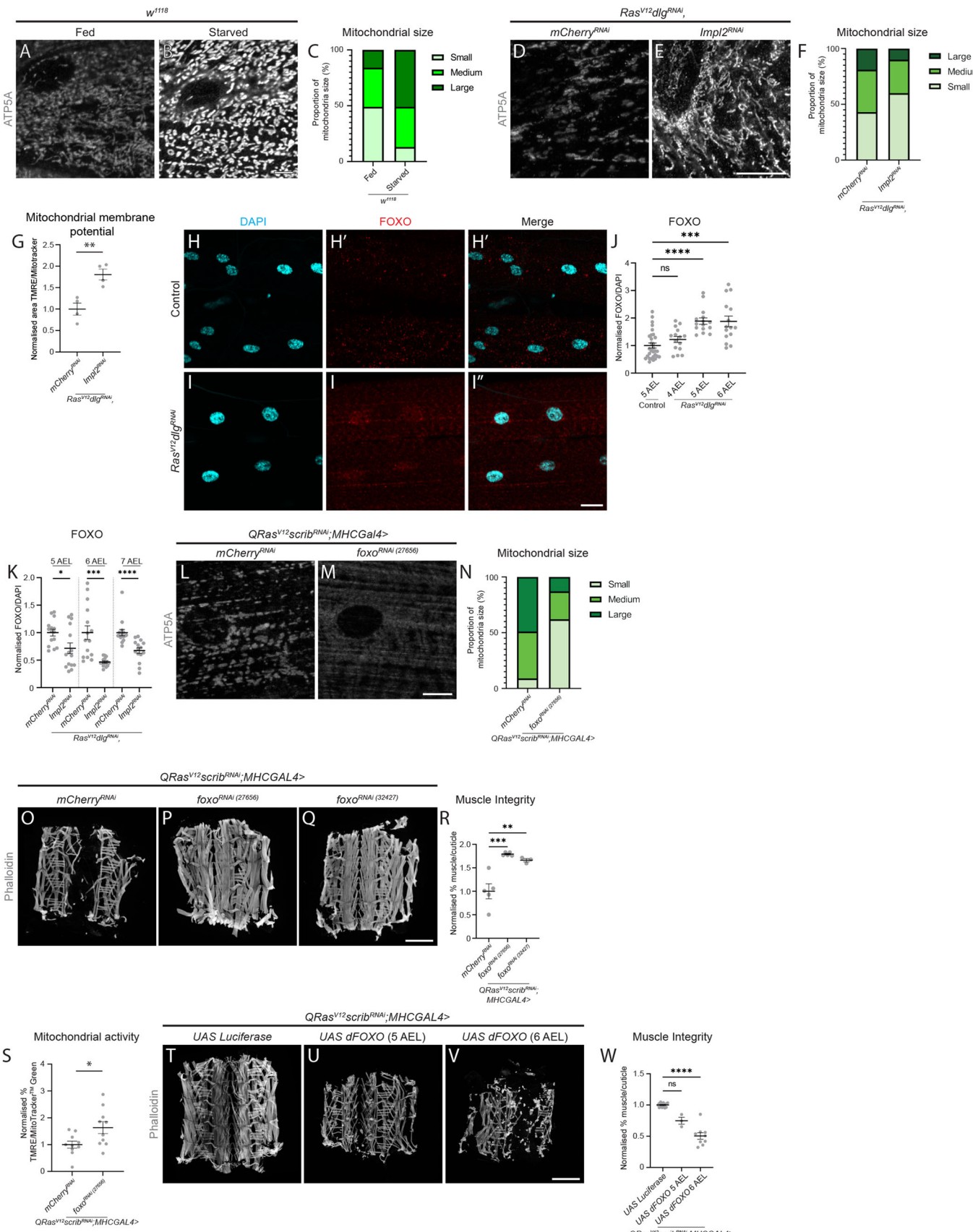

**Figure 2. Tumour-secreted Impl2 mediates muscle mitochondrial morphology via FOXO.**

(A, B) Representative images of ATP5A staining of mitochondria in the muscles of $w^{1118}$ larvae raised on a normal diet (Fed) and $w^{1118}$ larvae undergoing nutrient restriction (starved) from critical weight (60 h after larval hatching (ALH)) dissected at 5 days AEL. (C) Quantification of the proportion of small, medium, and large mitochondria as a percentage of total mitochondria in (A, B) performed using Chi-square test ($P < 0.0001$, $n = 3, 3$). (D, E) Representative images of ATP5A staining of mitochondria in the muscles of $Ras^{V12}dlg1^{RNAi},mCherry^{RNAi}$ and $Ras^{V12}dlg1^{RNAi},ImpL2^{RNAi}$ larvae (6 days AEL). (F) Quantification of the proportion of small, medium, and large mitochondria as a percentage of total mitochondria in (D, E) performed using Chi-square test ($P = 0.038$, $n = 3, 3$). (G) Quantification of the percentage of total mitochondria stained with MitoTracker™ Green that are shown to be active via TMRE incorporation in the muscles of 7 days AEL $Ras^{V12}dlg1^{RNAi},mCherry^{RNAi}$ and $Ras^{V12}dlg1^{RNAi},ImpL2^{RNAi}$ larvae, performed using Student's $t$ test ($n = 4, 4$). (H–I'') Representative images of control (5 days AEL) and $Ras^{V12}dlg1^{RNAi}$ larval muscles (7 days AEL) stained with DAPI to label the nucleus (H, I), and FOXO (H', I'). Merged images are shown in (H'', I''). (J) Quantification of nuclear FOXO staining in control (5 AEL) and $Ras^{V12}dlg1^{RNAi}$ larvae at (4–6 AEL performed using Kruskal–Wallis ($n = 30, 15, 15$). (K) Quantification of nuclear FOXO staining of $Ras^{V12}dlg1^{RNAi},mCherry^{RNAi}$ and $Ras^{V12}dlg1^{RNAi},ImpL2^{RNAi}$ larval muscles at 5, 6, and 7 days AEL, performed using Mann–Whitney $U$ (5 and 7 days AEL), and Welch's $t$ test (6 days AEL, $n = 15, 15, 15, 15, 15, 15$). (L, M) Representative images of ATP5A staining of mitochondria in the muscles of $QRas^{V12}scrib^{RNAi};MhcGAL4>mCherry^{RNAi}$ and $QRas^{V12}scrib^{RNAi};MhcGAL4>foxo^{RNAi\,(27656)}$ larvae (6 days AEL). (N) Quantification of the proportion of small, medium, and large mitochondria as a percentage of total mitochondria in (L, M) performed using Chi-square test ($P < 0.0001$, $n = 5, 5$). (O–Q) Representative muscle fillets from $QRas^{V12}scrib^{RNAi};MhcGAL4>mCherry^{RNAi}$ (O), $QRas^{V12}scrib^{RNAi};MhcGAL4>foxo^{RNAi\,(27656)}$ (P), and $QRas^{V12}scrib^{RNAi};MhcGAL4>foxo^{RNAi\,(32427)}$ (Q) larvae (7 days AEL), stained with Phalloidin to visualise actin. (R) Quantification of muscle integrity in (O–Q) performed using one-way ANOVA ($n = 5, 5, 3$). (S) Quantification of the percentage of total mitochondria stained with MitoTracker™ Green that are shown to be active via TMRE incorporation in the muscles of 6 days AEL $QRas^{V12}scrib^{RNAi};MhcGAL4>mCherry^{RNAi}$ and $QRas^{V12}scrib^{RNAi};MhcGAL4>foxo^{RNAi\,(27656)}$ larvae, performed using Student's $t$ test ($n = 10, 10$). (T–V) Representative muscle fillets from $QRas^{V12}scrib^{RNAi};MhcGAL4 > UAS\ Luciferase$ (T), $QRas^{V12}scrib^{RNAi};MhcGAL4 > UAS\ dFOXO$ (5 days AEL, U), and $QRas^{V12}scrib^{RNAi};MhcGAL4 > UAS\ dFOXO$ (6 days AEL, V) larvae, stained with Phalloidin to visualise actin. (W) Quantification of muscle integrity in (T–V) performed using Brown–Forsythe ($n = 14, 3, 9$). Scale bars: 10 μm for (A, B, D, E, L, M), 20 μm for (H, H', H'', I, I', I''), and 500 μm for (O, P, Q, T, U, V). Data information: All error bars are $+/-$ SEM. $P$ values are: ns (not significant), $P > 0.05$; *$P < 0.05$; **$P < 0.01$; ***$P < 0.001$; ****$P < 0.0001$. Source data are available online for this figure.

which has previously been shown to sufficiently increase protein translation in the muscle (Kim and O'Connor, 2021), was also not able to improve muscle integrity (Fig. EV3FF–HH). Together these data suggest that defects in autophagy and protein synthesis are not the primary cause of muscle wasting.

## Mitochondrial fusion promotes lipid utilisation in cachectic muscles

To further investigate the role of insulin-mediated mitochondrial fusion on muscle wasting, we next examined how these changes affected lipid and glycogen metabolism in the muscle. Cachexia has been associated with the rewiring of lipid metabolism (Joshi and Patel, 2022), and we have previously shown that there is a significant reduction in total lipid stores of cachectic animals (Lodge et al, 2021). Consistent with a reduction in overall lipid stores, we observed a significant reduction in the number of lipid droplets (LDs, Fig. 3A–C visualised by LipidTOX™), in the muscles of tumour-bearing animals ($Ras^{V12}dlg1^{RNAi}$). The depletion of lipids is phenocopied by nutrient restriction in wild-type animals (Fig. 3D–F). Furthermore, inhibition of ImpL2 in the tumour was sufficient to significantly increase muscle LD levels by 6 days AEL ($Ras^{V12}dlg1^{RNAi}$, Fig. 3G–I). Together, our data suggest that the depletion of muscle lipid stores in cachectic animals lies downstream of tumour-secreted ImpL2.

The lack of LDs in the muscle could be accounted for by an increase in lipolysis, whereby fat storage is broken down and released as free fatty acids. However, we found that by promoting LD formation through the overexpression of lipid storage droplet-2 (Lsd2, Fig. EV4A–C) did not prevent the loss of muscle integrity in tumour-bearing animals ($QRas^{V12}scrib^{RNAi}$, Fig. EV4D–F).

Next, we tested if inhibiting mitochondria fusion via Marf RNAi could restore lipid stores. We found that Marf knockdown in the muscles of tumour-bearing animals improved LD numbers ($QRas^{V12}scrib^{RNAi}$, Fig. 3J–L). It is known that utilisation of glycogen, which is a readily available source of energy, often occurs before fat breakdown (Soeters et al, 2012). Consistent with this, we found that glycogen stores in the muscles were depleted earlier than lipid

stores at 5 days AEL ($QRas^{V12}scrib^{RNAi}$, Fig. 3M–O). However, muscle-specific knockdown of Marf was not able to preserve the loss of muscle glycogen stores ($QRas^{V12}scrib^{RNAi}$, Fig. 3P–R), suggesting that mitochondrial fusion had greater effects on lipid rather than glycogen stores in the muscles of cachectic animals.

## Inhibition of muscle-specific beta-oxidation improves muscle integrity in cachectic animals

To better understand the metabolic changes that occur in the muscles of tumour-bearing animals, we conducted proteomics to assess for protein expression levels in wild-type and tumour-bearing animals ($Ras^{V12}dlg1^{RNAi}$, Fig. 4A,B). Consistent with the downregulation of Mhc and OPP levels (Fig. EV3Q–V), we detected a global downregulation of translational proteins (Fig. 4A). Furthermore, we found that proteins involved in the GO term "metabolism of lipids" were significantly upregulated (Fig. 4A,B). Interestingly, CPT1A/Whd, an important regulator of the beta-oxidation pathway (Strub et al, 2008), was found to be upregulated both at the transcriptional ($QRas^{V12}scrib^{RNAi}$, Fig. 4C) and protein levels ($Ras^{V12}dlg1^{RNAi}$, Fig. 4D) in the muscles of tumour-bearing animals. Beta-oxidation is the metabolic process by which fatty acids are broken down into acetyl-CoA molecules, which can then be used to produce energy (Fig. 4E). As beta-oxidation proceeds, the fatty acids stored in the LDs become depleted, leading to a reduction in the size and abundance of LDs in the cell. To test if preventing beta-oxidation could affect muscle integrity, we next knocked down Whd specifically in the muscles of tumour-bearing animals ($QRas^{V12}scrib^{RNAi}$). We found that Whd inhibition was sufficient to increase LD numbers (Fig. 4F) and improve muscle integrity (Fig. 4G–I), without altering tumour size (Fig. EV5A). Interestingly, while Whd knockdown was able to restore mitochondrial membrane potential (as assessed via TMRE, Fig. 4J), it was unable to restore mitochondrial size (Figs. 4K–M and EV5B) or muscle ATP (Fig. EV5C).

To assess if $whd$ is regulated by insulin signalling, we knocked down FOXO via RNAi in the muscle of tumour-bearing animals ($QRas^{V12}scrib^{RNAi}$, Fig. 4C), and found the $whd$ transcription was

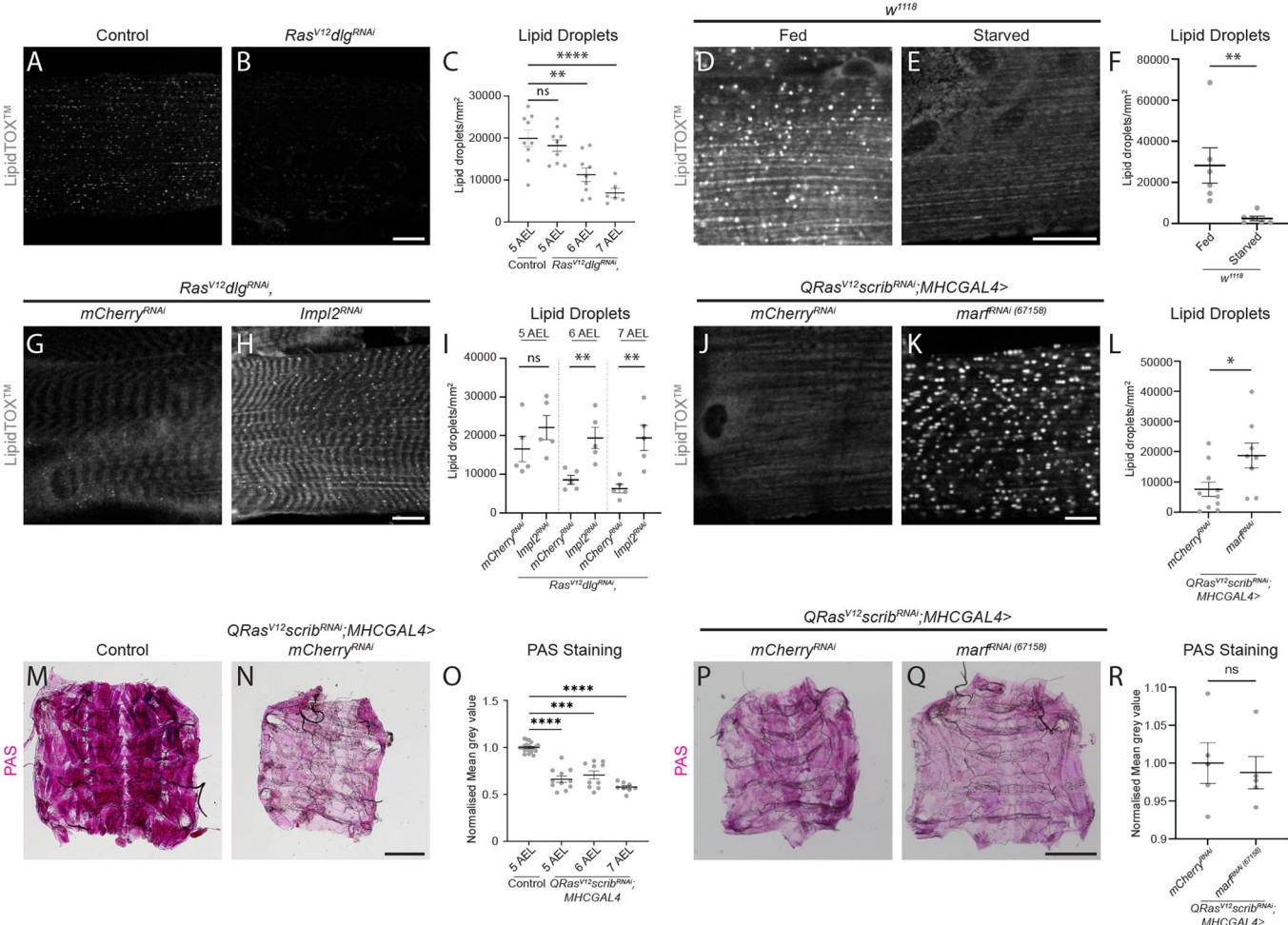

**Figure 3. Blocking mitochondrial fusion restores muscle lipid stores but not glycogen stores.**

(A, B) Lipid droplets (LDs) stained with LipidTOX™ in the muscles of control (5 days AEL) and $Ras^{V12}dlg1^{RNAi}$ larvae (7 days AEL). (C) Quantification of the number of LDs/ $mm^2$ in control (5 AEL) and $Ras^{V12}dlg1^{RNAi}$ larvae (5–7 AEL), performed using one-way ANOVA ($n = 9, 9, 9, 6$). (D, E) LDs stained with LipidTOX™ in the muscles of $w^{1118}$ larvae (5 days AEL) raised on a normal diet (Fed) and $w^{1118}$ larvae undergoing nutrient restriction (Starved, 5 days AEL) from critical weight (60 h after larval hatching (ALH)). (F) Quantification of the number of LDs/$mm^2$ in (D, E) performed using Mann–Whitney $U$ ($n = 6, 6$). (G, H) LDs stained with LipidTOX™ in the muscles of $Ras^{V12}dlg1^{RNAi},mCherry^{RNAi}$ and $Ras^{V12}dlg1^{RNAi},ImpL2^{RNAi}$ larvae (7 AEL). (I) Quantification of the number of LDs/$mm^2$ at 5–7 AEL performed using Student's $t$ tests ($n = 5, 5, 5, 5, 5, 5$). (J, K) LDs stained with LipidTOX™ in the muscles of $QRas^{V12}scrib^{RNAi};MhcGAL4>mCherry^{RNAi}$ and $QRas^{V12}scrib^{RNAi};MhcGAL4>marf^{RNAi (67158)}$ larvae (7 AEL). (L) Quantification of the number of LDs/$mm^2$ in (J, K) performed using Student's $t$ test ($n = 10, 8$). (M, N) Representative images of muscle fillets from control (5 AEL) and $QRas^{V12}scrib^{RNAi};MhcGAL4>mCherry^{RNAi}$ larvae (7 AEL), stained with periodic acid solution (PAS) to visualise glycogen. (O) Quantification of PAS staining in control and $QRas^{V12}scrib^{RNAi};MhcGAL4>mCherry^{RNAi}$ at days 5–7 AEL using Brown–Forsythe ($n = 21, 11, 10, 8$). (P, Q) Representative images of muscle fillets from $QRas^{V12}scrib^{RNAi};MhcGAL4>mCherry^{RNAi}$ and $QRas^{V12}scrib^{RNAi};MhcGAL4>marf^{RNAi (67158)}$ larvae (7 AEL), stained with periodic acid solution (PAS) to visualise polysaccharides such as glycogen. (R) Quantification of PAS staining in (P, Q) performed using Student's $t$ test ($n = 5, 5$). Scale bars: 20 μm for (A, B, D, E, G, H, J, K), and 500 μm for (M, N, P, Q). Data information: All error bars are +/− SEM. $P$ values are: ns (not significant), $P > 0.05$; *$P < 0.05$; **$P < 0.01$; ***$P < 0.001$; ****$P < 0.0001$. Source data are available online for this figure.

significantly downregulated. Together, these data suggest that FOXO lies upstream of Whd, a key regulator of beta-oxidation and mitochondrial membrane potential.

## Alterations in lipid metabolism is correlated with cachexia in the muscles of mouse and patient samples

To assess if there is an association between lipid metabolism and cachexia in patient samples, we examined the relationship between muscle *CPT1A* mRNA levels and muscle radiation attenuation (an indicator of muscle quality), using a microarray dataset from

patients with pancreatic ductal adenocarcinoma (PDAC) (Judge et al, 2020). We found a significant negative correlation of these two parameters (Fig. 4N), suggesting that increased *CPT1A* is correlated with poor muscle health in cachectic PDAC patients.

Next, we examined whether lipid metabolism plays a role in the Colon-26 (C-26) xenograft model in the mouse. Cross-sections through muscle samples of control vs. C-26 mice demonstrated a significant depletion of lipid stores, as indicated by a significant decrease in extramyocellular LD number and size (Fig. 4O–R, shaded in red). Interestingly, when we examined intramyocellular LD levels, we observed an increase in the density and size of LDs in

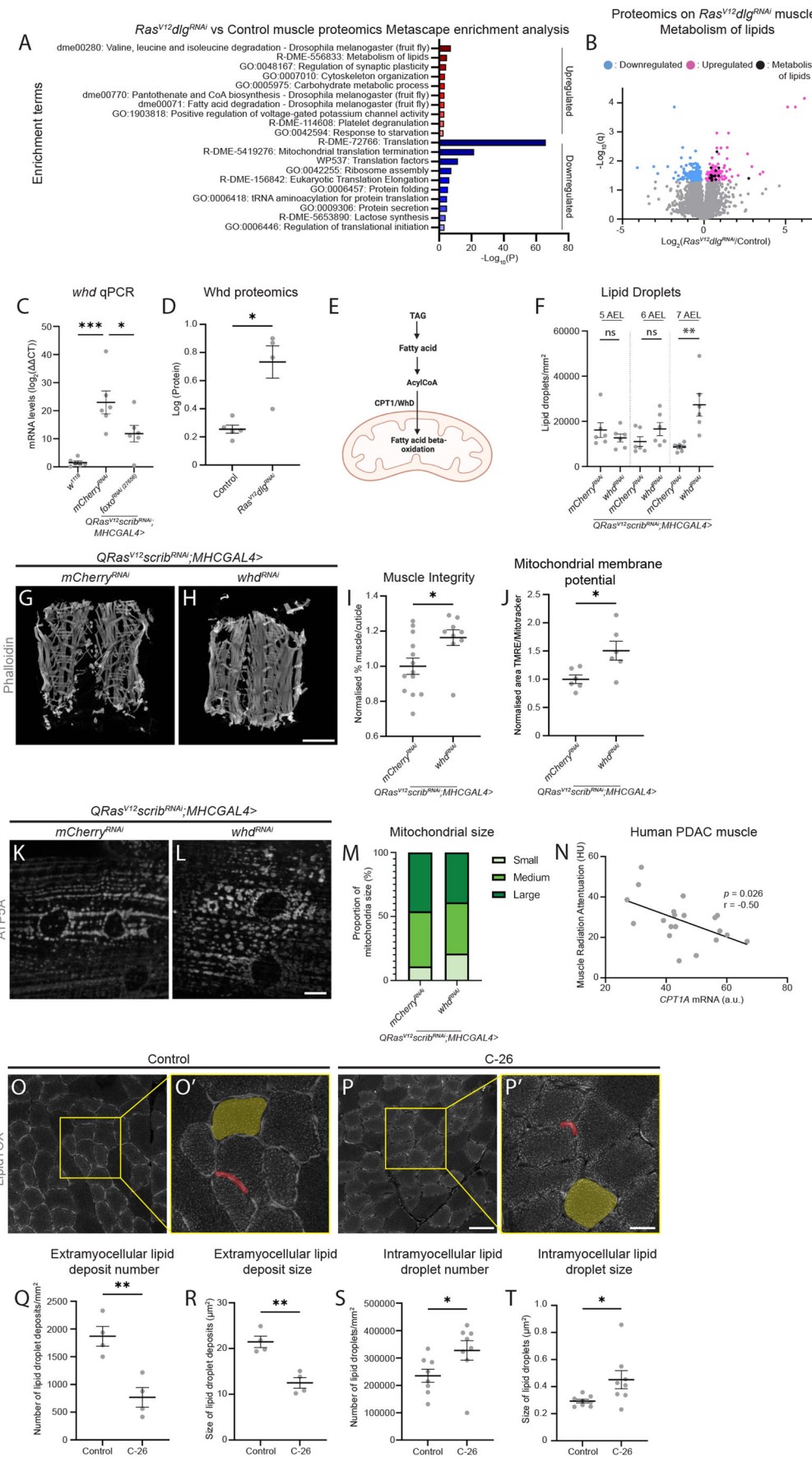

◄  **Figure 4.  Lipid metabolism via beta-oxidation is a mediator of muscle wasting.**

(A) Proteomics were performed using the muscles from 5 days AEL control and 7 days AEL *Ras^V12^dlg1^RNAi^* larvae (*n* = 5, 4). Differential expression analysis was performed using *t* tests with multiple hypothesis correction using Benjamini–Hochberg FDR adjustment. Pathway enrichment analysis was then performed using Metascape, separately for the significant up- and downregulated proteins. The top ten up- and downregulated enrichment terms from Metascape enrichment analysis are displayed. (B) Volcano plot showing differentially regulated proteins in control vs *Ras^V12^dlg1^RNAi^* larval muscles, with proteins found in the "R-DME-556833: Metabolism of lipids" enrichment term highlighted (*n* = 5, 4). (C) Quantification of qPCR results examining *whd* mRNA levels in *w^1118^* (5 days AEL) *QRas^V12^scrib^RNAi^;MhcGAL4>mCherry^RNAi^*, and *QRas^V12^scrib^RNAi^;MhcGAL4>foxo^RNAi (27656)^* larval muscles at 6 days AEL, performed using one-way ANOVA (*n* = 6, 6, 6). (D) Quantification of Log transformed Whd protein levels, from proteomics in (B), performed using Mann–Whitney *U* (*n* = 5, 4). (E) Schematic detailing Whd's role in mitochondrial beta-oxidation. (F) Quantification of the number of lipid droplets/mm² in *QRas^V12^scrib^RNAi^;MhcGAL4>mCherry^RNAi^* and *QRas^V12^scrib^RNAi^;MhcGAL4>whd^RNAi^* larval muscles, at 5, 6 and 7 days AEL, performed using Mann–Whitney *U* (5 and 6 days AEL) and Student's *t* test (7 days AEL, *n* = 6, 6, 6). (G, H) Representative muscle fillets from *QRas^V12^scrib^RNAi^;MhcGAL4>mCherry^RNAi^* and *QRas^V12^scrib^RNAi^;MhcGAL4>whd^RNAi^* larvae (7 days AEL), stained with Phalloidin to visualise actin. (I) Quantification of muscle integrity in (G, H) performed using Student's *t* test (*n* = 13, 9). (J) Quantification of the percentage of total mitochondria stained with MitoTracker™ Green that are shown to be active via TMRE incorporation in the muscles of 6 days AEL *QRas^V12^scrib^RNAi^;MhcGAL4>mCherry^RNAi^* and *QRas^V12^scrib^RNAi^;MhcGAL4>whd^RNAi^* larvae, performed using Student's *t* test (*n* = 6, 6). (K, L) Representative images of ATP5A staining of mitochondria in the muscles of *QRas^V12^scrib^RNAi^;MhcGAL4>mCherry^RNAi^* and *QRas^V12^scrib^RNAi^;MhcGAL4>whd^RNAi^* larvae (6 days AEL). (M) Quantification of the proportion of small, medium, and large mitochondria out of total mitochondria in (K, L) performed using Chi-square test (*P* = 0.15, *n* = 5, 5). (N) Quantification of human PDAC muscle *CPT1A* mRNA levels plotted against Muscle Radiation attenuation, performed using non-parametric Spearman's Correlation (*n* = 20). a.u. average unit. (O–P') Representative images of cross-sections of Type IIa muscle fibres in mouse tibiallis anterior muscle, stained with LipidTOX™, taken from control (O) and C-26 tumour (P) animals. Yellow boxes highlight zoomed in areas to better visualise intramyocellular and extramyocellular lipid deposits. Examples of extramyocellular lipid droplets have been labelled with red shading. Examples of where intramyocellular lipid droplets were measured have been labelled with yellow shading. The samples were harvested at 12 weeks plus 17–25 days after subcutaneous injection of PBS or C-26 tumour cells. (Q) Quantification of the number of extramyocellular lipid deposits/mm² in (O, P) performed using Student's *t* test (*n* = 4, 4). (R) Quantification of the size of extramyocellular lipid deposits in (O, P) performed using Student's *t* test (*n* = 4, 4). (S) Quantification of the number of intramyocellular lipid deposits/mm² in (O, P) performed using Student's *t* test (*n* = 4, 4). (T) Quantification of the size of intramyocellular lipid deposits in (O, P) performed using Student's *t* test (*n* = 4, 4). Scale bars: 10 µm for (K, L), 20 µm for (O', P'), 50 µm for (O, P), and 500 µm for (G, H). Data information: All error bars are +/− SEM. *P* values are: ns (not significant), *P* > 0.05; *P < 0.05; **P < 0.01; ***P < 0.001. Source data are available online for this figure.

cachectic muscles compared to control (Fig. 4O–P',S,T, shaded in yellow), which is consistent with reports of intramyocellular LD accumulation reported in human cachectic muscles (Stephens et al, 2011; Weber et al, 2009). Together, our data suggest that muscle lipid localisation and storage is also relevant in mouse models of cancer cachexia.

## Modulation via dietary supplementation of nicotinamide or a high-fat diet improves muscle integrity in cachectic animals

To determine whether we could prevent mitochondrial damage and muscle disruption in tumour-bearing animals through dietary modulation, we fed cachectic animals with diets supplemented with nicotinamide or coconut oil. Nicotinamide is a precursor of Vitamin B3 and is known to improve mitochondrial health (Jia et al, 2008). Tumour-bearing animals (*QRas^V12^scrib^RNAi^*) fed a diet containing 1 g/kg of nicotinamide exhibited reduced muscle mitochondrial size (Figs. 5A–C and EV5D) without causing changes in tumour size (Fig. EV5E) or mitochondrial membrane potential (Fig. EV5F). However, this diet was sufficient to cause increased muscle ATP levels (Fig. EV5G, *P* = 0.09), decreased FOXO levels (Fig. 5D), and improved muscle integrity (Fig. 5E–G). Consistent with our previous observation that FOXO RNAi was sufficient to reduce *whd* transcript levels (Fig. 4C), we found that nicotinamide supplementation also resulted in a significant reduction of *whd* (Fig. 5H). Together, our data indicate that nicotinamide feeding helps to improve muscle integrity in cachectic animals through the modulation of mitochondrial size, insulin signalling and *whd* levels, resulting in improved mitochondria membrane potential and the ability to convert cellular energy.

Coconut oil has previously been shown to enhance overall lipid levels in *Drosophila* larvae (Birse et al, 2010). We found this high-fat diet significantly improved LD levels in the muscles of tumour-

bearing animals (*Ras^V12^dlg1^RNAi^*, Fig. 5I–K). This dietary supplementation significantly improved muscle integrity (Fig. 5L–N), mitochondrial membrane potential (Fig. 5O) and ATP levels (Fig. 5P) without affecting the size of the tumour (Fig. EV5H). We also observed a small decrease in FOXO levels (Fig. 5Q), however, this did not result in changes in the level of *whd* transcription (Fig. 5R). This suggests that coconut oil supplementation, while modulates LD, mitochondrial membrane potential, ATP levels and insulin signalling, does not affect beta-oxidation per se.

Together, these data demonstrate that dietary supplementation of nicotinamide or lipids can potentially improve outcomes in cachexia through the modulation of insulin signalling, lipid metabolism or beta-oxidation.

## Discussion

Utilising a *Drosophila* model of cachexia, we have demonstrated that muscle wasting in cachexia is mediated by two main mechanisms: insulin signalling via FOXO and mitochondrial beta-oxidation (Fig. 5S). First, tumour-secreted ImpL2 systemically decreases insulin signalling, which results in an increase in FOXO nuclear localisation in the muscle. This is accompanied by an increase in mitochondrial size and a depletion of the muscle lipid stores. Knockdown of tumour-secreted ImpL2, or muscle-specific FOXO inhibition, resulted in reduced mitochondrial size and improved muscle integrity. Furthermore, inhibition of mitochondrial fusion via Marf, or beta-oxidation via Whd, were also able to improve mitochondrial membrane potential, muscle lipid stores and overall muscle integrity. Finally, supplementing dietary nicotinamide or lipids through a high-fat diet was sufficient to enhance ATP levels and muscle integrity.

Changes in mitochondrial morphology have been reported in the muscles of cachectic patients, however, so far, the underlying

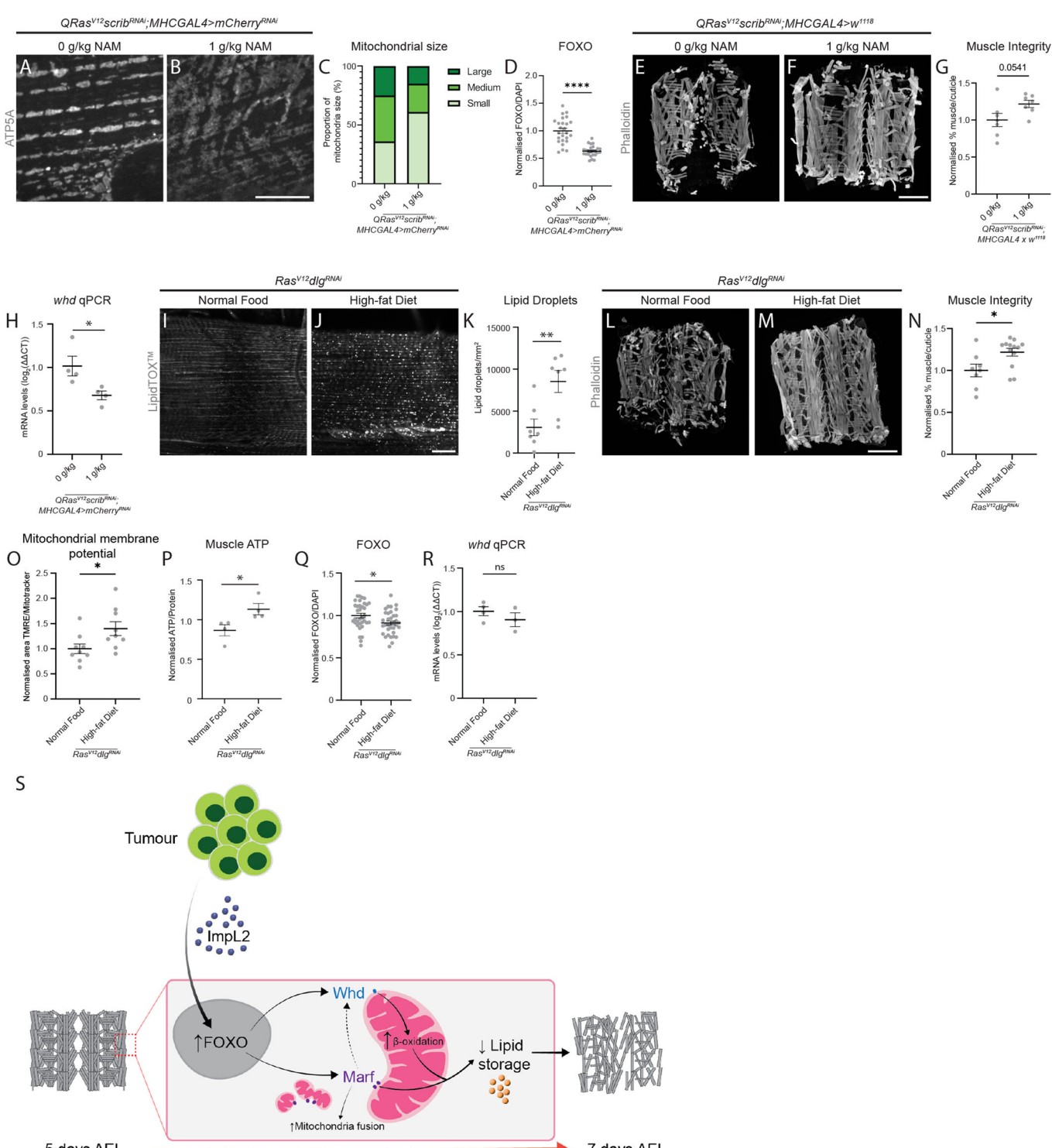

mechanisms are not clear. In our study, we have shown that mitochondrial fusion, beta-oxidation and the muscle lipid store are closely linked. It is likely that increased mitochondria fusion leads to a depletion of fatty acid stores in cachectic muscle, which contributes to muscle damage. Interestingly, in the muscles of mice and humans, it has been reported that fatty acid beta-oxidation and

mitochondrial dynamics can influence each other (Castro-Sepulveda et al, 2020; Jheng et al, 2012; Liu et al, 2014). Consistent with this, we also demonstrate a negative correlation between CTP1 levels and poorer outcomes in cachectic patients (Fig. 4N).

It has been reported that cachectic patients exhibit decreased dietary lipids and increased fatty acid release from the adipose

**Figure 5. Nicotinamide and high-fat diets improve muscle health in tumour-bearing animals.**

(A, B) Representative images of ATP5A staining of mitochondria in the muscles of $QRas^{V12}scrib^{RNAi}$;$MhcGAL4$>$mCherry^{RNAi}$ larvae (7 AEL) raised on a normal diet, or a diet containing 1 g/kg nicotinamide (NAM). (C) Quantification of the proportion of small, medium, and large mitochondria as a percentage of total mitochondria in (A, B) performed using Chi-square test ($P = 0.0019$, $n = 5$, 5). (D) Quantification of nuclear FOXO staining in the muscles of 6 days AEL $QRas^{V12}scrib^{RNAi}$;$MhcGAL4$>$mCherry^{RNAi}$ larvae raised on a normal diet, or a diet containing 1 g/kg NAM, performed using Welch's $t$ test ($n = 25$, 25). (E, F) Representative muscle fillets from $QRas^{V12}scrib^{RNAi}$;$MhcGAL4 > w^{1118}$ larvae raised on a normal diet, or a diet containing 1 g/kg NAM (7 AEL), stained with Phalloidin to visualise actin. (G) Quantification of muscle integrity in (E, F) performed using Student's $t$ test ($n = 7$, 7). (H) Quantification of qPCR results examining $whd$ mRNA levels in 6 days AEL $QRas^{V12}scrib^{RNAi}$;$MhcGAL4$>$mCherry^{RNAi}$ larvae raised on a normal diet, or a diet containing 1 g/kg NAM, performed using Student's $t$ test ($n = 4$, 4). (I, J) Lipid droplets (LDs) stained with LipidTOX™ in the muscles of $Ras^{V12}dlg1^{RNAi}$ larvae (6 days AEL) raised on a normal diet, or a high-fat diet. (K) Quantification of the number of LDs/mm² in (H, I) performed using Mann–Whitney $U$ ($n = 7$, 7). (L, M) Representative muscle fillets from $Ras^{V12}dlg1^{RNAi}$ larvae (7 AEL) raised on a normal diet, or a high-fat diet, stained with Phalloidin to visualise actin. (N) Quantification of muscle integrity in (L, M) performed using Mann–Whitney $U$ ($n = 8$, 13). (O) Quantification of the percentage of total mitochondria stained with MitoTracker™ Green that are shown to be active via TMRE incorporation in the muscles of 6 days AEL $Ras^{V12}dlg1^{RNAi}$ larvae raised on a normal diet, or raised on a high-fat diet, performed using Student's $t$ test ($n = 9$, 9). (P) Quantification of normalised muscle ATP for 7 days AEL $Ras^{V12}dlg1^{RNAi}$ larvae raised on a normal diet, or a high fat diet, performed using Student's $t$ test ($n = 4$, 4). (Q) Quantification of nuclear FOXO staining in the muscles of 7 days AEL $QRas^{V12}scrib^{RNAi}$;$MhcGAL4$>$mCherry^{RNAi}$ larvae raised on a normal diet, or a high-fat diet, performed using Student's $t$ test ($n = 35$, 35). (R) Quantification of qPCR results examining $whd$ mRNA levels in 7 days AEL $Ras^{V12}dlg1^{RNAi}$ larvae raised on a normal diet, or a high-fat diet, performed using Student's $t$ test ($n = 4$, 4). (S) Tumour-secreted ImpL2 mediates insulin signalling in the muscle of tumour-bearing animals by influencing the nuclear localisation of FOXO. This reduction in muscle insulin signalling influences mitochondria fusion via Marf. Increased mitochondrial fusion is accompanied by a decrease in muscle lipid stores, increased fatty acid beta-oxidation via Whd in the mitochondria, and a reduction in mitochondria membrane potential, contributing to a loss of muscle integrity in cachectic animals. Scale bars: 10 µm for (A, B), 20 µm for (I, J), and 500 µm for (E, F, L, M). Data information: All error bars are $+/-$ SEM. $P$ values are: ns (not significant), $P > 0.05$; *$P < 0.05$; **$P < 0.01$; ****$P < 0.0001$. Source data are available online for this figure.

tissue (Hodson et al, 2008). It is therefore likely that the depletion of available energy substrates (glycogen and fatty acids) contributes to mitochondrial dysfunction and loss of muscle integrity. In this study, we have shown that coconut oil supplementation increases mitochondrial function and improves muscle integrity. While the underlying mechanism is so far unclear, we think this dietary supplementation most likely acts to increase lipid stores in general, therefore delaying mitochondrial damage caused by beta-oxidation. However, it is also possible that a high-fat diet could first lead to reduced mitochondrial size (Jheng et al, 2012; Liu et al, 2014), which in turn can cause decreased beta-oxidation (Castro-Sepulveda et al, 2020), therefore, leading to a slower utilisation of lipids.

We reported a decrease in LD number in the muscles of both cachectic flies and mice. In contrast with this, studies in cachectic patient muscle samples have previously reported an accumulation of intramyocellular lipids (Stephens et al, 2011; Weber et al, 2009). When we examined LD levels in the flies, we did not measure intramyocellular lipid levels, as our images were taken longitudinally. Upon examination of LD levels in the muscles of the C-26 mouse model (via transverse section), we saw a decrease in extramyocellular LD levels, and an increase in intramyocellular LDs. It is therefore possible that the depletion of LDs we observed is in fact a mobilisation of extramyocellular/subsarcolemmal LDs to intramyocellular LDs, for their use in mitochondrial beta-oxidation. This would be consistent with reports that in starved mouse embryonic fibroblasts, fatty acids stored in LDs undergo transfer to mitochondria for beta-oxidation (Rambold et al, 2015). In the future, it would be interesting to explore whether the increase in intramyocellular LDs in human cachexia muscle samples also correlates with a decrease in extramyocellular LDs, and if this is associated with changes in fatty acid beta-oxidation.

Together, our study demonstrates that targeting FOXO, mitochondrial fusion/beta-oxidation and replenishing lipid stores in cachectic animals can prevent muscle wasting in cachexia. Therefore, this work opens up new avenues for finding therapeutic targets to prevent or attenuate the progression of cancer cachexia in patients.

# Methods

## *Drosophila* stocks and husbandry

The following stocks were used from the Bloomington *Drosophila* stock centre:

*Mef2GAL4* (BL27391),

*MhcGAL4* (BL55133),

*PhmGAL4*

*UAS-Cat.A* (BL24621)

*UAS Drp1* (BL51647),

*UAS-foxo^{RNAi}* (BL27656),

*UAS-marf* (BL67157),

*UAS-foxo^{RNAi}* (BL32427), (out of the two RNAi's, $foxo^{RNAi\ (27656)}$ showed the stronger effect and was used for all other experiments

Torso RNAi ((Rewitz et al, 2009))

*UAS luciferase* (BL64774),

*UAS-marf^{RNAi}* (BL31157),

*UAS-marf^{RNAi}* (BL67158) (of the two RNAi's, $marf^{RNAi\ (67158)}$ showed the stronger effect and was used for all the experiments)

*UAS-mCherry^{RNAi}* (BL35785),

*UAS-Opa1^{RNAi}* (BL32358),

*UAS pGFP-mCherryAtg8a* (BL37749),

*UAS-S6K^{CA}* (BL6914),

*UAS Sod1* (BL24750),

*UAS-whd^{RNAi}* (BL33635),

*ey-FLP1;act > CD2 > GAL4,UAS-GFP* (Lodge et al, 2021).

The following stocks were obtained from the Vienna Drosophila Resource Centre:

*UAS-Impl2^{RNAi}* (v30931).

The following stocks were also used:

$w^{1118}$,

*ey-FLP1;UAS-dlg1 RNAi,UAS-RasV12 /CyO, Gal80;act > CD2 > GAL4,UAS-GFP,*

*Ey-FLP1; QUAS-Ras^{V12}, QUAS-scrib^{RNAi}/ CyOQS; MhcGal4, act > CD2 > QF, UAS-RFP/TMBQS* (Lodge et al, 2021),

*Ey-FLP1; QUAS-Ras^{V12}, QUAS-scrib^{RNAi}/ CyOQS; Mef2Gal4, act > CD2 > QF, UAS-RFP/TMBQS* (Lodge et al, 2021),

*UAS-Atg1^{RNAi}* (Donna Denton),
*UAS-Atg1 6A* (Donna Denton),
*UAS dFOXO* (Kieran Harvey),
*UAS-GFP* (Kieran Harvey),
*UAS GPx1* (Tatsushi Igaki),
*UAS-lacZ^{RNAi}* (Kieran Harvey),
*UAS lsd2* (Alex Gould),
*UAS-Ras^{V12}* (Helena Richardson),
*mCherryAtg8a/CyOGFP,*
*mCherryAtg8a/CyOGFP;mCherry^{RNAi}/TM6cSb,*
*mCherryAtg8a/CyOGFP;ImpL2^{RNAi}/TM6cSb.*

Fly stocks were reared on standard *Drosophila* media, adults were allowed to lay for 24 h at 25 °C and the progeny was then moved to 29 °C. Experiments were conducted on animals lacking tumours at the wandering stage, and on tumour-bearing animals on a specific number of days after egg lay as indicated throughout the methods.

## Mitochondrial analysis

The automated analysis of mitochondrial cross-sectional areas was conducted on z-stacks from confocal image stacks modified from (Dhyani, 2024). First, the background noise was identified using the *imsegkmeans* function in MATLAB, which segregated the pixels into two categories: background and mitochondria based on the contrast in pixel intensity, allowing an unbiased removal of background noise.

Subsequently, each image was converted into a feature matrix, comprising three key features: the *x* and *y* coordinates, and the fluorescent intensity of each pixel. This feature matrix was then used for DBSCAN (Density-Based Spatial Clustering of Applications with Noise)-based clustering, a technique that implements image segmentation by employing feature-based pixel clustering. Notably, DBSCAN does not require prior knowledge of the quantity or dimensions of the mitochondria in the image. The DBSCAN-based segmentation process requires only two key parameters: the minimum cluster (or mitochondrion) size and the search radius. These parameters were fine-tuned using Monte Carlo simulations on a representative image from each experiment. The objective was to achieve an Intersection over Union (IOU) accuracy above 95% between the segmented output and manual annotation of the representative image. The derived parameters were then consistently applied across all subsequent images from the experiment. Finally, the sizes of the mitochondria were ascertained by calculating the sum of pixels in each cluster, further multiplied by the cluster resolution.

## Tumour size assessment

Tumour-bearing larvae were heat fixed for 3–5 s in 65 °C water and imaged within 24 h using Nikon SoRA spinning disk confocal using the ×4 objective. Analysis was carried out using Volocity software through the measurement of object intensity.

## Dietary supplementation and nutrient restriction

Fly food supplemented with nicotinamide at a concentration of 1 g of nicotinamide per 1 kg of normal food was created as follows. In all, 100 g of regular food was melted in a beaker using a microwave until there were no solid lumps. Overall, 100 mg of nicotinamide (Jia et al, 2008) (Sigma-Aldrich, #N0636-100G) was dissolved in 1 ml of sterile water at room temperature, added to the melted fly food, and mixed thoroughly. The combined mixture was divided into ten fly vials with ~10 ml of food per vial. In total, 100 g of regular food without nicotinamide was also melted and poured into ten vials to be used as control food. Both nicotinamide and regular food was left to cool at 4 °C overnight. To control for density in feeding, crosses were laid on regular food. On day 1 AEL, before the embryos hatched, 20–30 embryos were taken and placed on a piece of cardboard and placed into the control or nicotinamide food. The larvae that hatch on the food were then fed on the specified diets until dissection.

High-fat diet fly food was created as follows. Overall, 70 g of regular food was melted in a beaker using a microwave until there were no solid lumps. In total, 30 g of coconut oil (Birse et al, 2010) (Community Co Virgin Coconut Oil 450 ml) was melted in a separate beaker using a microwave until there were no solid lumps. The melted coconut oil was then poured into the beaker with the 70 g regular food and mixed thoroughly. The mixture was then left to cool slightly for 5–10 min, then was mixed thoroughly again to prevent the food from separating from the coconut oil. This was repeated if the mixture appeared to separate again. The combined mixture was divided into ten fly vials with ~10 ml of food per vial. In all, 100 g of regular food without coconut oil was also melted and poured into ten vials to be used as control food. Both high-fat food and regular food was left to cool at 4 °C overnight. A small piece of tissue paper (~1 × 3 cm) was placed into the food of control and high-fat vials to soak up excess coconut oil. To control for density in feeding, crosses were laid on regular food. On day 1 AEL, before the embryos hatched, 20–30 embryos were taken and placed on a piece of cardboard and placed into the control or high-fat food. The larvae that hatch on the food were then fed on the specified diets until dissection.

For starvation/nutrient restriction, the larvae were fed on normal food for 60 h. Approximately 20 larvae were transferred onto either normal food or a diet consisting of 1% agar in PBS at this time point. The animals were then either fed or starved for 24 h before dissection.

### *Drosophila* immunostaining

For FOXO, Mhc staining as well as phalloidin staining (muscle integrity), larvae were heat-killed (Lodge et al, 2021; Dark et al, 2022), muscle fillets (prepared as previously described (Lodge et al, 2021; Dark et al, 2022)) were then fixed for 20 min in PBS containing 4% formaldehyde and washed three times for 10 min each with PBS containing 0.3% Triton-X (PBST-0.3). For Atg8a and LipidTOX™ experiments, animals were dissected in cold 1× PBS, fixed for 45 min and washes were performed in 1× PBS. For ATP5A staining, DHE and TMRE experiments, animals were dissected in cold *Drosophila* Schneider's Medium. ATP5A samples were fixed for 20 min followed by three 10 min washes in PBST-0.3. For DHE and TMRE there was no fixation or wash steps before staining. Tissues were then stained as per the manufacturer's specifications. All samples were imaged on an Olympus FV3000 confocal microscope. Muscle integrity and TMRE samples were imaged using a ×10 objective lens. FOXO, Mhc, Atg8a, LipidTOX™, and DHE were imaged with a ×40 objective lens. ATP5A was imaged with a combination of ×40 and ×63 lenses using the

FV3000 or the Nikon SoRA spinning disk confocal (60x objective for the Nicotinamide ATP5A experiment). All confocal samples were mounted in glycerol except for LipidTOX™ and TMRE, which were mounted in 1× PBS and 25 nM TMRE in *Drosophila* Schneider's Medium, respectively. Within a given experiment, all images were acquired using identical settings. Primary antibodies used: dFOXO (Abcam, 1:100, #ab195977), Mhc (DHSB, 1:10, #3E8-3D3), ATP5A (Abcam, 1:500, #ab14748), Tiggrin (Zhang and Cadigan, 2017) (rabbit, 1:50), Secondary donkey antibodies conjugated to Alexa 488 and Alexa 555 (Molecular Probes) were used at 1:200. DAPI 405 (Abcam, #ab228549) was used at 1:10,000, Phalloidin 647 (Abcam, #ab176759) was used at 1:10,000, HCS LipidTOX™ Red Neutral Lipid Stain (Invitrogen, #H34476) was used at 1:1000. MitoTracker™ Green FM (Invitrogen, #M7514) was used at a concentration of 250 nm (Parker et al, 2016). TMRE was used as previously described (Invitrogen, #T669, 100 nm) (Rana et al, 2017). DHE staining was performed as previously described (10 min DHE stain) (Owusu-Ansah et al, 2008). To assay for translation in the muscle, we used Click-iT Plus OPP Alexa Fluor 488 Protein Synthesis Assay Kit (Thermo Fisher, #C10456) (Lodge et al, 2021).

## Glycogen staining

Muscle fillets were dissected in 1% BSA in PBS, fixed in 4% formaldehyde in PBS for 20 min, and washed twice in 1% BSA in PBS. Periodic acid stain (PAS) was used as previously described (Yamada et al, 2018). The samples were mounted in glycerol and imaged on an Olympus BX53 Brightfield microscope using a ×4 objective lens.

## ATP assay

ATP assay was conducted according to Tennessen et al, 2014, we used five muscle fillets which were dissected in cold PBS. 10 µl of the supernatant was transferred into a 1.5-ml microfuge tube and dilute 1:10 with 90 µl dilution buffer [25 mM Tris (pH 7.8), 100 µM EDTA], then transfer 10 µl of the diluted supernatant to a second 1.5-ml tube and dilute 1:20 with the dilution buffer. ATP kit used was from Molecular Probes (A22066), and Bradford assay was from Biorad (5000002).

## Experimental animals (mice)

All experiments were approved by the Animal Ethics Committee of The University of Melbourne and conducted in accordance with the Australian Code of Practice for the care and use of animals for scientific purposes as stipulated by the National Health and Medical Research Council (Australia). Male Balb/c mice were obtained from the Animal Resources Centre (Canning Vale, Western Australia). All mice were housed in the Biological Research Facility under a 12:12 h light–dark cycle, with water and standard laboratory chow available ad libitum.

## Mouse model of cancer cachexia

The procedures used to thaw and count the Colon-26 (C-26) cells used to inject mice has been previously described (Murphy et al, 2012). Twelve-week-old male Balb/c mice were anaesthetised with isoflurane (induction, 3–4% oxygen-isoflurane at 0.5 L min$^{-1}$;

maintenance, 2–3% at 0.5 L min$^{-1}$), such that they were unresponsive to tactile stimuli. Mice were then given a subcutaneous (s.c.) injection of $5 \times 10^5$ C-26 cells suspended in 100 µl of sterilised phosphate-buffered saline (PBS; $n = 4$) or 100 µl of sterilised PBS only (control; $n = 4$) and recovered from anaesthesia on a heat pad. After 17–25 days, when end-point criteria was met, mice were anaesthetized deeply with sodium pentobarbitone (Nembutal; 60 mg/kg; Sigma-Aldrich, Castle Hill, NSW, Australia) via intra-peritoneal (i.p.) injection and the tibialis anterior (TA) muscles were carefully excised, blotted on filter paper, trimmed of tendons and any adhering non-muscle tissue and weighed on an analytical balance. The LTA muscle was mounted in embedding medium, frozen in thawing isopentane and stored at −80 °C for subsequent analyses. Mice were killed by cardiac excision while still anaesthetised deeply.

## Mouse immunostaining

Serial sections (8 µm) were cut transversely through the TA muscle using a refrigerated (−20 °C) cryostat (CTI Cryostat; IEC, Needham Heights, MA, USA). Frozen mouse TA tissue was thawed at room temperature for 10 min, then fixed in 4% neutral buffered formalin for 10 min and washed twice for 5 min in 1× PBS. Samples were then stained as per the manufacturer's specifications. Samples were mounted in 1× PBS and imaged on an Olympus FV3000 confocal microscope with a ×40 objective lens. All images were acquired using identical settings. HCS LipidTOX™ Red Neutral Lipid Stain (Invitrogen, #H34476) was used at 1:1000.

## Image analysis

All images were quantified using FIJI (Schindelin et al, 2012). FOXO intensity was normalised to DAPI, FOXO and DAPI levels were quantified by drawing a circle around the nucleus in the DAPI channel, and the mean grey value (m.g.v.) was determined for FOXO and DAPI channels. To measure fluorescence intensity of Mhc, OPP, Tiggrin, Atg8a, PAS and DHE, a ROI was drawn around a sarcomere (Mhc), muscle junction (Tiggrin), nucleus plus adjacent to nucleus (Atg8a), section of a muscle segment (PAS), and nucleus only (DHE), on the z-plane where fluorescence was most intense. For the Mhc, OPP, Tiggrin, single Atg8a, PAS, and DHE quantifications, the levels of fluorescence were calculated with respect to background fluorescence, using total corrected cell fluorescence (TCCF), as described previously (McCloy et al, 2014). For the Tandem Atg8a-mCherry-GFP quantifications, a measurement was taken in both the mCherry and GFP channels, and total autophagy was calculated as a ratio of Atg8a-mCherry to Atg8a-GFP.

Percentage muscle/cuticle was determined using FIJI as previously described (Lodge et al, 2021; Dark et al, 2022). In brief, dissected muscle fillets stained with Phalloidin to mark actin were analysed using a FIJI macro (Dark et al, 2022). A ROI was drawn around the cuticle of the muscle fillet, and the image was converted to a binary mask using the "Auto Threshold" tool. The total area of fluorescence detected within the ROI was divided by the total ROI area, which we calculated as % muscle attachment. The value of control is represented as "1" and the experimental data normalised to the control.

The number of LDs present in the fly muscle was determined through the use of a macro in FIJI (Computer Code EV1). In brief, files were imported into FIJI, and a 200 × 200 pixel ROI was created

on an 8-bit converted representative slice. The image was cropped to the ROI, then the "Auto-threshold" function was used to convert the image into binary. The "Analyse Particles" function using a size range of "0.00–10" was then used to count the number of LDs present in the image. This was then normalised to the size of the ROI in mm$^2$.

The number of large extramyocellular LDs present in the mouse muscle was determined using a macro in FIJI (Computer Code EV1). In brief, files were imported into FIJI, a representative slice was converted to 8-bit. The "Auto-threshold" function was used to convert the image into binary. The "Analyse Particles" function using a size range of "2-infinity" was then used to count the number and size of extramyocellular LDs present in the image. The number of LDs was then normalised to the size of the ROI in mm$^2$.

The number of intramyocellular LDs present in the mouse muscle was determined using a macro in FIJI (Computer Code EV1). In brief, files were imported into FIJI, and the "Auto-threshold" function was used to convert the image into binary. Five polygon ROIs that each encompassed the interior of a different myofiber was created on an 8-bit converted representative slice. The "Analyse Particles" function using a size range of "0.00–10" was then used to count the number and size of LDs present in the image. The number and size of LDs were averaged between the five myofibers for one image. The number of LDs was then normalised to the size of the ROI in mm$^2$.

The proportion of mitochondrial sizes in the muscle was determined through applying a $Log_{10}$ transformation to the list of areas outputted from the automated mitochondrial analysis pipeline (see above) to bring it closer to a normal distribution. The data were then binned into three sizes, those with a $Log_{10}$ transformed value: $X \leq -0.5$ (small), $-0.5 < X \leq 0$ (medium), and $X > 0$ (large). The percentage of mitochondria in each of these categories was averaged across replicates, and the distribution of mitochondrial size across categories was compared via Chi-square test.

The level of TMRE membrane potential was determined through the use of a macro in FIJI (Computer Code EV2). In brief, files were imported into FIJI and split into MitoTracker™ Green and TMRE channels. A max intensity z-projection of the MitoTracker™ Green channel was converted to 8-bit, smoothed using the "Gaussian Blur" and "Remove Outliers" functions, and a square ROI was drawn around the muscle fillet. The image was then converted to a binary mask using the "Auto Threshold" function using the "IsoData" method. A selection was then created around all the thresholded MitoTracker™ Green area, and this area was measured. A max intensity z-projection of the TMRE channel was then processed in the same way as the MitoTracker™ Green channel. The final result was an area measurement for total mitochondria via MitoTracker™ Green and active mitochondria via TMRE. The total area of TMRE was divided by the total area of MitoTracker™ Green, to give a percentage value of how many total mitochondria are active.

## Electron microscopy

*Drosophila* 3rd instar larvae were dissected and fixed in 2.5% glutaraldehyde solution in 0.1 M sodium cacodylate buffer overnight at 4 °C or for 2 h at room temperature. The samples were then washed with 0.1 M sodium cacodylate, followed by staining with 1% osmium tetroxide and 2% uranyl acetate using a Pelco Biowave and washed in 0.1 M sodium cacodylate. They were then dehydrated in an ethanol series (1×–50%, 70%, 90% and 2×–100%) followed by infiltration with increasing concentrations of epon resin (1×–25%, 50%, 75% and 2×–100%). Samples were subsequently processed in the resin with the Biowave high vacuum function before being embedded in fresh resin and polymerised in a 60 °C oven for 48 h. Formvar-coated, one-slot grids were used to collect thin sections (50 nm) obtained via a Leica Ultracut UC6 Ultramicrotome by taking longitudinal sections of muscles. Images were collected in a JEOL 1011 electron microscope at 80 kV.

## Proteomics

Samples were lysed by tip-probe sonication in 1% SDS containing 10 mM tris(2-carboxyethyl)phosphine and 40 mM chloroacetamide in 100 mM HEPES pH 8.5. The lysate was incubated at 95 °C for 5 min and centrifuged at 20,000×*g* for 30 min at 4 °C. Peptides were prepared using a modified SP3 approach with paramagnetic beads (Hughes et al, 2019). Briefly, lysates were shaken with a 1:1 mixture of hydrophilic and hydrophobic Sera-Mag SpeedBeads (GE Healthcare) in a final concentration of 50% ethanol for 8 min at 23 °C. The beads were washed three times with 80% ethanol and dried at 23 °C for 20 min. Proteins were digested directly on the beads in 50 μL of 10% trifluoroethanol in 100 mM HEPES, pH 7.5 sequencing-grade LysC (Wako Chemicals) and sequencing-grade trypsin (Sigma) for 16 h at 37 °C. The supernatant-containing peptides were removed and mixed with 150 μl of 1% trifluoroacetic acid (TFA) and purified using styrenedivinylbenzene- reverse phase sulfonate microcolumns. The columns were washed with 100 μl of 99% isopropanol containing 1% TFA followed by 100 μl of 99% ethyl acetate containing 1% TFA followed by 5% acetonitrile containing 0.2% TFA and eluted with 80% acetonitrile containing 1% ammonium hydroxide then dried by vacuum centrifugation. Peptides were resuspended 2% acetonitrile, 0.1% TFA and stored at −80 °C.

Peptides were separated on a 40 cm × 75 μm inner diameter PepMap column packed with 1.9 μm C18 particles (Thermo Fisher) using Dionex nanoUHPLC. Peptides were separated using a linear gradient of 5–30% Buffer B over 70 min at 300 nl/min (Buffer A = 0.1% formic acid; Buffer B = 80% acetonitrile, 0.1% formic acid). The column was maintained at 50 °C coupled directly to an Orbitrap Exploris 480 mass spectrometer (MS). A full-scan MS1 was measured at 60,000 resolution at 200 *m/z* (350–951 *m/z*; 50 ms injection time; 2.5e6 automatic gain control target) followed by data-independent analysis (16 *m/z* isolation with 37 windows and a 1 *m/z* overlap, 28 normalised collision energy; 30 K resolution; auto-injection time, 2e6 automatic gain control target).

Mass spectrometry data were processed using Spectronaut DirectDIA (v15.1.210713.50606) and searched against the Drosophila melanogaster UniProt database (October 2019) using all default settings with peptide spectral matches and protein false discovery rate (FDR) set to 1%. The data were searched with a maximum of 2 miss-cleavages, and methionine oxidation and protein N-terminus acetylation were set as variable modifications while carbamidomethylation of cysteine was set as a fixed modification. Quantification was performed using MS2-based

extracted ion chromatograms employing 3–6 fragment ions >450 *m/z* with automated fragment-ion interference removal as described previously (Bruderer et al, 2015). Data was analysed in Perseus(Tyanova et al, 2016) and included median normalisation and differential expression analysis using *t* tests with multiple hypothesis correction using Benjamini–Hochberg FDR adjustment.

Pathway enrichment analysis was performed for both the statistically significant differentially expressed up- and down-regulated proteins, using the express analysis function in Metascape ([http://metascape.org], Version 3.5, February 26, 2023) (Zhou et al, 2019).

### Human sample collection

Collection of biospecimens of rectus abdominus muscle from patients with pathologically diagnosed pancreatic ductal adenocarcinoma (PDAC; *n* = 10 females/10 males) undergoing tumour resection surgery was compliant with an approved Institutional Review Board protocol at the University of Florida, with written informed consent obtained from all patients, and conformed to the Declaration of Helsinki. The detailed patient demographics and analysis of preoperative CT scans for skeletal muscle index (SMI) and muscle radiation attenuation as quantitative measures of skeletal muscularity and myosteatosis, respectively, have been published previously (Judge et al, 2020, 2019). The methods for obtaining the microarray data have been published previously.

### Statistical analysis

All statistical analyses were conducted using GraphPad Prism 9.0 (©GraphPad software Inc.). For experiments measuring % muscle/cuticle, FOXO/DAPI levels, PAS staining, Mhc staining, DHE staining, Atg8a intensity, and TMRE/MitoTracker™ Green ratios, experimental values were normalised to the average value of their respective controls. At least three animals per genotype were used for all muscle experiments. For FOXO, DHE, Atg8a, Mhc staining intensity quantifications, individual data points represent the fluorescence intensity of a single nucleus or muscle fibre. For muscle integrity, PAS staining, LD number, mitochondrial size, and TMRE quantifications, individual data points represent a single larva. For experiments with two genotypes or treatments, two-tailed unpaired student's *t* tests were used to test for significant differences. The Welch's correction was applied in cases of unequal variances, and the Mann–Whitney *U* tests was used in the cases of violated normality. For experiments with more than two genotypes, significant differences between specific genotypes were tested using a one-way ANOVA and a subsequent Šidák post hoc test. A Brown–Forsythe correction was applied in cases of unequal variances, and in the cases of violated normality, the Kruskal–Wallis test was used. The results for all post hoc tests conducted in a given analysis are shown in graphs. For all graphs, error bars represent SEM. *P* and adjusted-*P* values are reported as follows: $P > 0.05$, ns (not significant); $*P < 0.05$; $**P < 0.01$; $***P < 0.001$; $****P < 0.0001$.

## Data availability

This study includes no data deposited in external repositories.

## Peer review information

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

## Acknowledgements

We are grateful to Kieran Harvey, Donna Denton, Helena Richardson, Alex Gould, and Tatsushi Igaki for generous sharing of fly stocks and antibodies. We would like to thank Bloomington Drosophila Stock Center, Vienna Drosophila Resource Center and Developmental Studies Hybridoma Bank for fly stocks and antibodies. We would also like to thank OZDros for *Drosophila* quarantine, Peter MacCallum Cancer Institute Microscopy Core for technical assistance. We are grateful to Kellie Veen and Khanh Nguyen for assistance with illustrations, Yuchen Bai for technical assistance with mouse immunostaining experiments, Michelle Meier for assistance with mitochondrial quantifications and Daniel Bakopolous for intellectual input in this project. We would also like to thank Edel Alvarez and Khanh Nguyen for the critical reading of the manuscript. This work is funded by the NHMRC Ideas Grant APP1182847 and the Peter MacCallum Cancer Foundation.

## Author contributions

**Callum Dark**: Conceptualization; Resources; Data curation; Formal analysis; Supervision; Validation; Investigation; Visualisation; Methodology; Writing—original draft; Project administration; Writing—review and editing. **Nashia Ali**: Conceptualisation; Data curation; Formal analysis; Investigation; Methodology. **Sofya Golenkina**: Conceptualisation; Resources; Data curation; Software; Formal analysis; Validation; Investigation; Visualisation. **Vaibhav Dhyani**: Data curation; Formal analysis. **Ronnie Blazev**: Data curation; Formal analysis. **Benjamin L Parker**: Data curation; Formal analysis. **Kate T Murphy**: Resources; Methodology. **Gordon S Lynch**: Supervision. **Tarosi Senapati**: Formal analysis; Methodology. **S Sean Millard**: Resources; Supervision. **Sarah M Judge**: Resources. **Andrew R Judge**: Resources. **Lopamudra Giri**: Resources; Supervision. **Sarah M Russell**: Resources; Supervision. **Louise Y Cheng**: Conceptualisation; Resources; Data curation; Formal analysis; Supervision; Funding acquisition; Validation; Investigation; Methodology; Writing—original draft; Project administration; Writing—review and editing.

## Disclosure and competing interests statement

The authors declare no competing interests.

# Expanded View Figures

**Figure EV1. Mitochondria size/membrane potential in *QRas^V12^scrib^RNAi^*, manipulation of developmental delay does not alter mitochondria size or membrane potential and ROS manipulations in the muscles of tumour-bearing animals.** ▶

(A) Size distribution of individual mitochondria in the muscles of control (5 AEL) and *Ras^V12^dlg1^RNAi^* larvae (4–7 AEL) ($n = 1351$, 1950, 1313, 1285, 1214). (B) Proportion of small, medium, and large mitochondria as a percentage of total mitochondria in control (5 AEL) and *QRas^V12^scrib^RNAi^;MhcGal4* muscles (4–7 AEL), performed using Chi-square test ($P = 0.72$, $n = 3$, 3, 3, 3, 3). (C) Size distribution of individual mitochondria in the muscles of control (5 AEL) and *QRas^V12^scrib^RNAi^;MhcGal4* muscles (4–7 AEL) ($n = 427$, 609, 841, 589, 717). (D–E″) Zoomed in images of control (5 AEL) and *Ras^V12^dlg1^RNAi^* (7 AEL) larval muscles stained with MitoTracker™ Green which labels all mitochondria, and active mitochondria with TMRE (zoomed out images are in Fig. 1J–K″). White arrows indicate an example of a mitochondrion that has no TMRE membrane potential. (F-G″) Representative images of control (5 days AEL) and *QRas^V12^scrib^RNAi^;MhcGAL4* (6 days AEL) larval muscle fillets stained with MitoTracker™ Green which labels all mitochondria (F, G), and active mitochondria with TMRE (F′, G′). (H) Quantification of the percentage of total mitochondria stained with MitoTracker™ Green that are also positive for TMRE in control (5 AEL) and *QRas^V12^scrib^RNAi^;MhcGAL4* (6 AEL) performed using Student's *t* test ($n = 5$, 5). (I) Quantification of normalised muscle ATP measured in control (5 AEL) and *QRas^V12^scrib^RNAi^;MhcGAL4* (6 AEL), performed using Student's *t* test ($n = 3$, 3). (J) Proportion of small, medium, and large mitochondria as a percentage of total mitochondria in control (5 AEL) and *phmGAL4>torso^RNAi^* (7 AEL) animals, performed using Chi-square test ($P > 0.99$, $n = 4$, 5). (K) Size distribution of individual mitochondria in the muscles of control (5 AEL) and *phmGAL4>torso^RNAi^* (7 AEL) animals ($n = 2815$, 2959). (L) Quantification of the percentage of total mitochondria stained with MitoTracker™ Green that are shown to be active via TMRE incorporation in the muscles of control (5 AEL) and *phmGAL4>torso^RNAi^* (7 AEL), performed using Student's *t* test ($n = 4$, 4). (M, N) Representative images of DHE staining in the muscles of control (6 AEL) and *QRas^V12^scrib^RNAi^;MhcGAL4* (6 AEL). (O) Quantification of DHE staining in M-N, performed using Mann–Whitney *U* ($n = 45$, 15).
(P, Q) Representative muscle fillets from *QRas^V12^scrib^RNAi^;Mef2GAL4>lacz^RNAi^;mCherry^RNAi^* and *QRas^V12^scrib^RNAi^;Mef2GAL4 > UAS CatalaseA;UAS Sod1* larvae (both 7 AEL), stained with Phalloidin to visualise actin. (R) Quantification of muscle integrity in (P, Q) performed using Student's *t* test ($n = 13$, 11). (S, T) Representative muscle fillets from *QRas^V12^scrib^RNAi^;MhcGAL4>mCherry^RNAi^*, *QRas^V12^scrib^RNAi^;MhcGAL4 > UAS GPx1* (7 days AEL) stained with Phalloidin to visualise actin. This data was part of an experiment with EV2 G-H and EV3 FF–GG, which use the same controls. (U) Quantification of muscle integrity in (S, T), performed using Kruskal–Wallis as part of an analysis with EV2 I and EV3 HH, which use the same controls ($n = 13$, 16). Scale bars: 10 μm for (D, D′, D″, E, E′, E″), 20 μm for (M, N), and 500 μm for (F, F′, F″, G, G′, G″, P, Q, S, T). Data information: All error bars are $+/-$ SEM. *P* values are: ns (not significant), $P < 0.05$; **$P < 0.001$; ****$P < 0.0001$. Source data are available online for this figure.

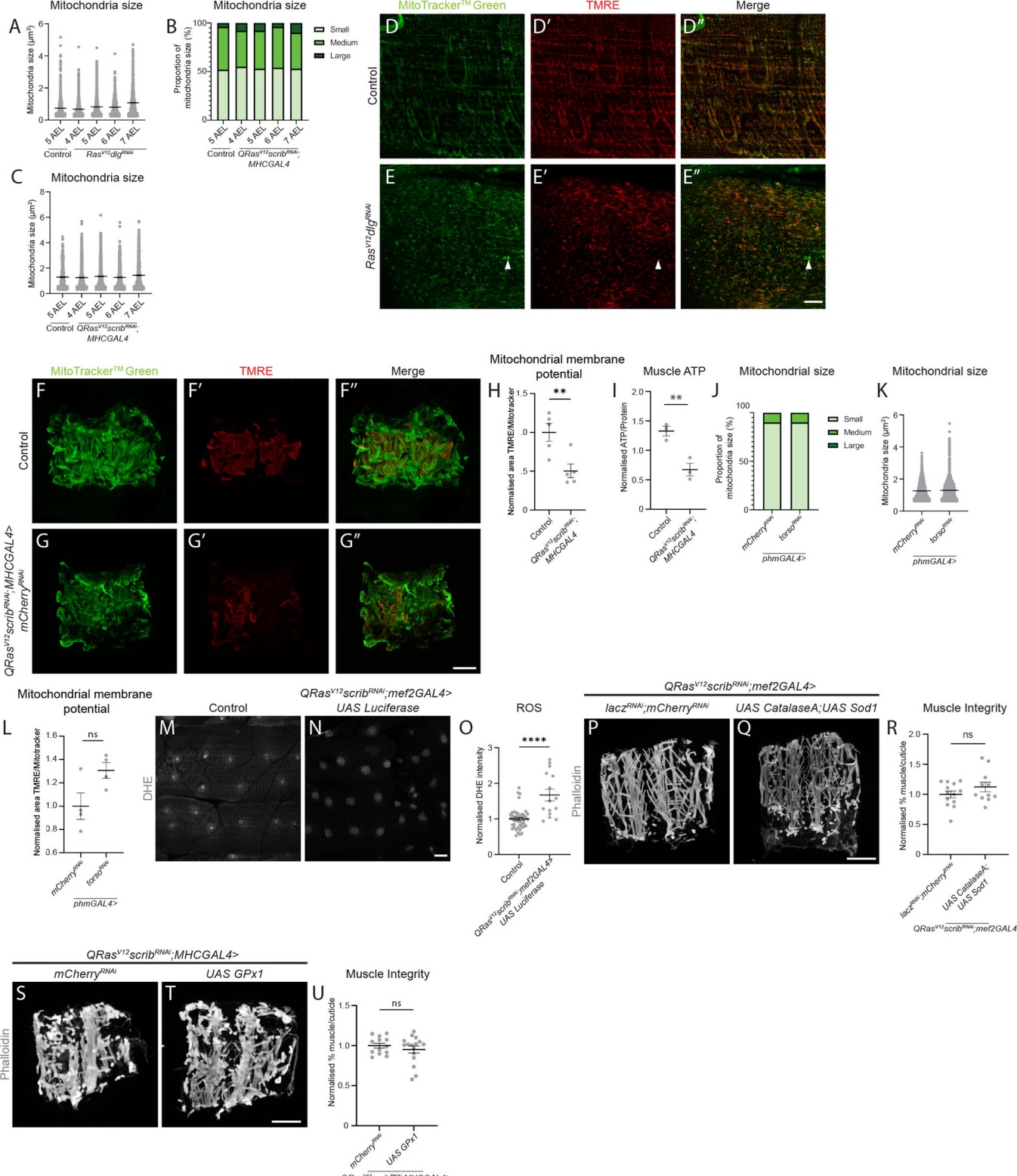

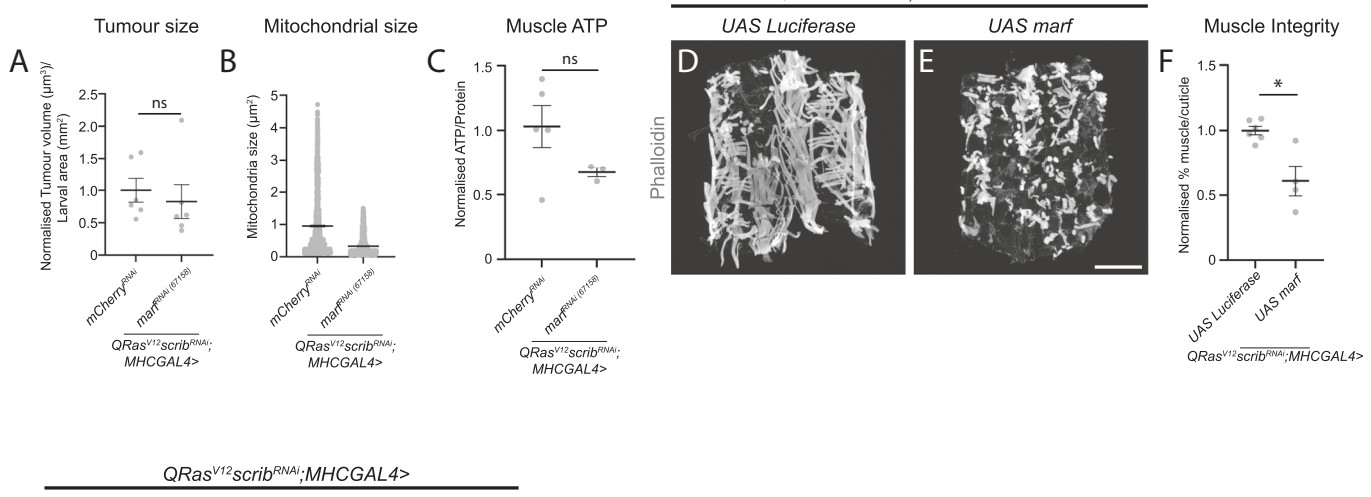

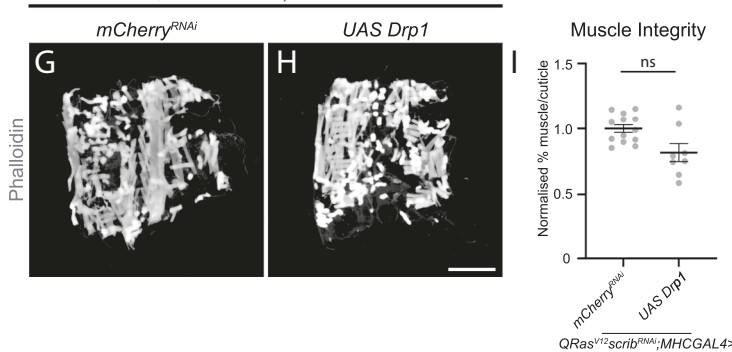

**Figure EV2.  Muscle mitochondria size, integrity, ATP tumour animals with muscle-specific marf and Drp1 overexpression compared to control.**

(A) Quantification of normalised tumour volume of $QRas^{V12}scrib^{RNAi};MhcGAL4>mCherry^{RNAi}$ and $QRas^{V12}scrib^{RNAi};MhcGAL4>marf^{RNAi\ (67I58)}$ larvae (7 AEL), performed using Kruskal–Wallis as part of an analysis with EV3 F and EV5 A, which use the same controls ($n = 6, 6$). (B) Size distribution of individual mitochondria in the muscles of $QRas^{V12}scrib^{RNAi};MhcGAL4>mCherry^{RNAi}$ and $QRas^{V12}scrib^{RNAi};MhcGAL4>marf^{RNAi\ (67I58)}$ larvae (6 AEL) ($n = 1516, 1690$). (C) Quantification of normalised muscle ATP in $QRas^{V12}scrib^{RNAi};MhcGAL4>mCherry^{RNAi}$ and $QRas^{V12}scrib^{RNAi};MhcGAL4>marf^{RNAi\ (67I58)}$ larvae (6 AEL), performed using one-way ANOVA as part of an analysis with EV3 E and EV5 B, which use the same controls ($n = 5, 3$). (D, E) Representative images of muscle fillets of $QRas^{V12}scrib^{RNAi};MhcGAL4 > UAS\ Luciferase$ and $QRas^{V12}scrib^{RNAi};MhcGAL4 > UAS\text{-}marf$ larvae (7 AEL). (F) Quantification of muscle integrity of $QRas^{V12}scrib^{RNAi};MhcGAL4 > UAS\ Luciferase$ and $QRas^{V12}scrib^{RNAi};MhcGAL4 > UAS\text{-}marf$ larvae (7 AEL), performed using Welch's $t$ test ($n = 6, 4$). (G, H) Representative images of muscle fillets of $QRas^{V12}scrib^{RNAi};MhcGAL4>mCherry^{RNAi}$ and $QRas^{V12}scrib^{RNAi};MhcGAL4 > UAS\ Drp1$ (7 AEL). This data was part of an experiment with EV1 S–T and EV3 FF–GG, which use the same controls. (I) Quantification of muscle integrity of $QRas^{V12}scrib^{RNAi};MhcGAL4>mCherry^{RNAi}$ and $QRas^{V12}scrib^{RNAi};MhcGAL4 > UAS\ Drp1$ (7 AEL), performed using Kruskal–Wallis as part of an analysis with EV1 U and EV3 HH, which use the same controls ($n = 13, 8$). Scale bars: 500 μm for (D, E, G, H). Data information: All error bars are $+/-$ SEM. $P$ values are: ns (not significant), *$P < 0.05$. Source data are available online for this figure.

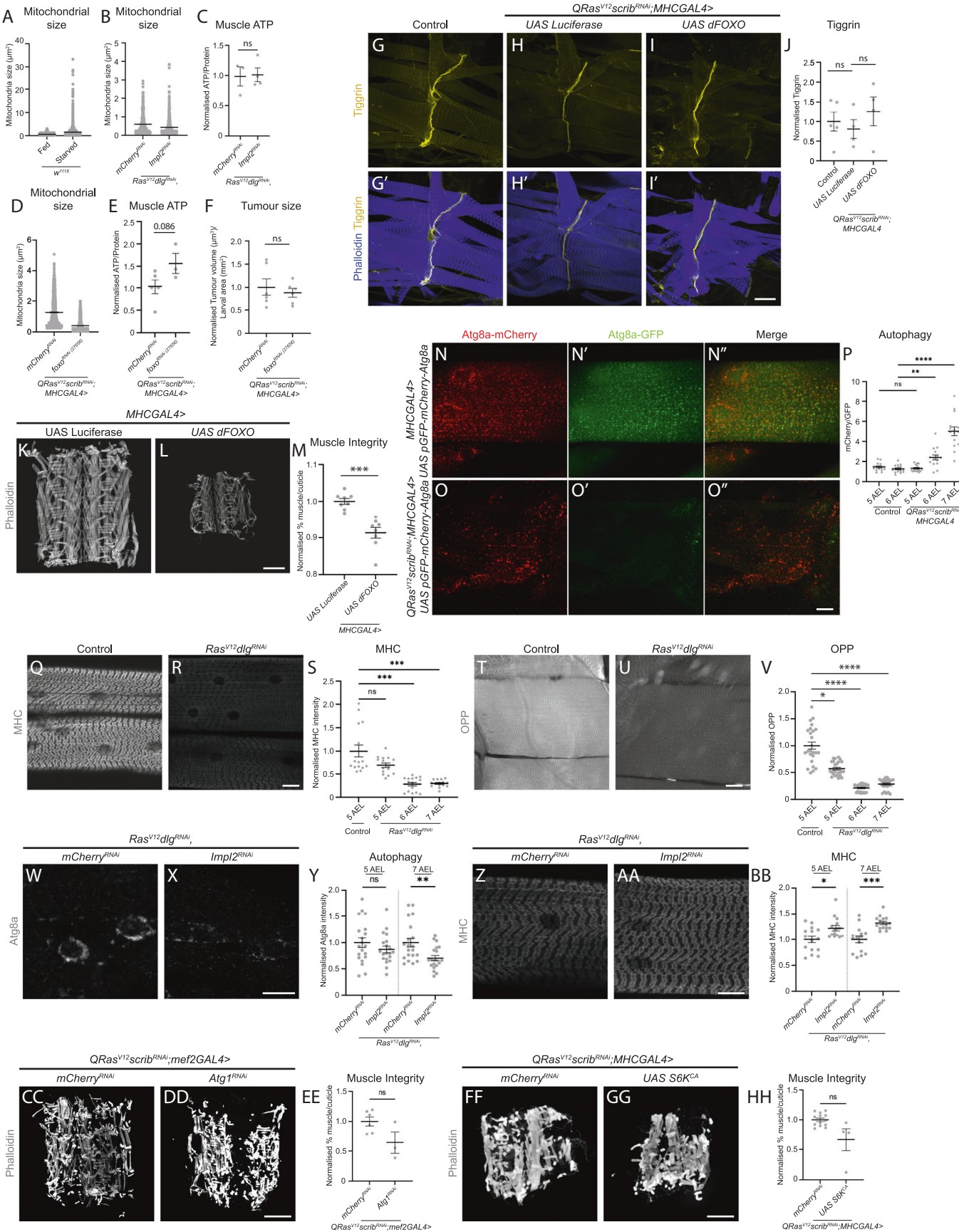

◀ **Figure EV3. Assessment of mitochondrial size, muscle ATP, tumour size, autophagy and translation under various genetic manipulations.**

(A) Size distribution of individual mitochondria in the muscles of $w^{1118}$ larvae raised on a normal diet (Fed) and $w^{1118}$ larvae undergoing nutrient restriction (Starved) from critical weight (60 h after larval hatching (ALH)) (5 AEL, n = 2248, 3747). (B) Size distribution of individual mitochondria in the muscles of $Ras^{V12}dlg1^{RNAi},mCherry^{RNAi}$ and $Ras^{V12}dlg1^{RNAi},ImpL2^{RNAi}$ larvae (7 AEL, n = 432, 1037). (C) Quantification of normalised muscle ATP of 7 AEL $Ras^{V12}dlg1^{RNAi},mCherry^{RNAi}$ and $Ras^{V12}dlg1^{RNAi},ImpL2^{RNAi}$ larvae, performed using Student's $t$ test (n = 3, 4). (D) Size distribution of individual mitochondria in the muscles of $QRas^{V12}scrib^{RNAi};MhcGAL4>mCherry^{RNAi}$, and $QRas^{V12}scrib^{RNAi};MhcGAL4>foxo^{RNAi\ (27656)}$ larvae (6 AEL, n = 1083, 2230). (E) Quantification of normalised muscle ATP of 6 AEL $QRas^{V12}scrib^{RNAi};MhcGAL4>mCherry^{RNAi}$, and $QRas^{V12}scrib^{RNAi};MhcGAL4>foxo^{RNAi\ (27656)}$ larvae, performed using one-way ANOVA as part of an analysis with EV2 C and EV5 B, which use the same controls (n = 5, 3). (F) Quantification of normalised tumour volume of 7 AEL $QRas^{V12}scrib^{RNAi};MhcGAL4>mCherry^{RNAi}$, and $QRas^{V12}scrib^{RNAi};MhcGAL4>foxo^{RNAi\ (27656)}$ larvae, performed using Kruskal–Wallis as part of an analysis with EV2 A and EV5 A, which use the same controls (n = 6, 6). (G-I') Representative images of Tiggrin staining marking a muscle/tendon junction in control (5 AEL, **G**, **G'**), $QRas^{V12}scrib^{RNAi};MhcGAL4 > UAS\ Luciferase$ (5 AEL, **H**, **H'**) and $QRas^{V12}scrib^{RNAi};MhcGAL4 > UAS\ dFOXO$ (5 AEL, **I**, **I'**) larval muscle. (J) Quantification of Tiggrin levels in control (5 AEL), $QRas^{V12}scrib^{RNAi};MhcGAL4 > UAS\ Luciferase$ and $QRas^{V12}scrib^{RNAi};MhcGAL4 > UAS\ dFOXO$ (5 AEL) larval muscles, performed using one-way ANOVA (n = 5, 4, 4). (K, L) Representative images of muscle fillets from $MhcGAL4 > UAS\ Luciferase$, $MhcGAL4 > UAS\ dFOXO$ (both 5 AEL). (M) Quantification of muscle integrity of $MhcGAL4 > UAS\ Luciferase$ and $MhcGAL4 > UAS\ dFOXO$ (5 AEL), performed using Student's $t$ test, (n = 8, 8). (N-O'') Representative images of larval muscles of $MhcGAL4$ (5 AEL).and $QRas^{V12}scrib^{RNAi};MhcGAL4$ animals (7 AEL) crossed to a reporter of autophagy, Atg8a, tagged with both mCherry (**N**, **O**) and GFP (**N'**,**O'**). Merged images are shown in (**N''**, **O''**). (P) Quantification of the ratio of Atg8a-mCherry to Atg8a-GFP in control and $Ras^{V12}dlg1^{RNAi}$ larvae at days 5–7 AEL, performed using Brown–Forsythe (n = 15, 15, 15, 15). (Q, R) Representative images of Myosin Heavy chain (Mhc) staining in the muscles of control (5 AEL) and $Ras^{V12}dlg1^{RNAi}$ (7 AEL) larvae. (S) Quantification of Mhc staining in control and $Ras^{V12}dlg1^{RNAi}$ larvae from days 5–7 AEL, performed using Brown–Forsythe (n = 15, 15, 15, 15). (T, U) Representative images of OPP staining in the muscles of control (5 AEL) and $Ras^{V12}dlg1^{RNAi}$ (7 AEL) larvae. (V) Quantification of OPP staining in the muscles of control (5 AEL) and $Ras^{V12}dlg1^{RNAi}$ (7 AEL) larvae, performed using Kruskal–Wallis (n = 25, 25, 25, 25). (W, X) Representative images of muscles of $Ras^{V12}dlg1^{RNAi},mCherry^{RNAi}$ (7 AEL) and $Ras^{V12}dlg1^{RNAi},ImpL2^{RNAi}$ (7 AEL) larvae crossed to a reporter of autophagy, Atg8a, tagged with mCherry. (Y) Quantification of Atg8a-mCherry levels in (**W**, **X**), as well as from earlier timepoints, performed using Student's $t$ test (5 days AEL), Welch's $t$ test (6 days AEL), and Mann–Whitney $U$ (7 days AEL, n = 20, 20, 20, 20, 20, 20). (Z, AA) Representative images of Myosin Heavy chain (Mhc) staining in the muscles of $Ras^{V12}dlg1^{RNAi},mCherry^{RNAi}$ (7 AEL) and $Ras^{V12}dlg1^{RNAi},ImpL2^{RNAi}$ (7 AEL) larvae. (BB) Quantification of Mhc staining in (**Z**, **AA**), as well as staining from earlier timepoints, performed using Mann–Whitney U (5 and 6 days AEL) and Welch's $t$ test (7 days AEL, n = 15, 15, 15, 15, 15, 15). (CC, DD) Representative muscle fillets from $QRas^{V12}scrib^{RNAi};Mef2GAL4>mCherry^{RNAi}$ and $QRas^{V12}scrib^{RNAi};Mef2GAL4>Atg1^{RNAi}$ larvae (both 7 AEL), stained with Phalloidin to visualise actin. (EE) Quantification of muscle integrity in (**CC**, **DD**) performed using Student's $t$ test (n = 6, 3). (FF, GG) Representative muscle fillets from $QRas^{V12}scrib^{RNAi};MhcGAL4>mCherry^{RNAi}$, $QRas^{V12}scrib^{RNAi};MhcGAL4 > UAS\text{-}S6K^{CA}$ larvae (both 7 AEL), stained with Phalloidin to visualise actin. This data was part of an experiment with EV1 S–T and EV2 G-H, which use the same controls. (HH) Quantification of muscle integrity in (**FF**, **GG**) performed using Kruskal–Wallis as part of an analysis with EV1 U and EV2 I, which used the same controls (n = 13, 4). Scale bars: 20 µm for (**N**, **N'**, **N''**, **O**, **O'**, **O''**, **Q**, **R**, **T**, **U**, **W**, **X**, **Z**, **AA**), 50 µm for (**G**, **G'**, **H**, **H'**, **I**, **I'**), and 500 µm for (**K**, **L**, **CC**, **DD**, **FF**, **GG**). Data information: All error bars are $+/-$ SEM. $P$ values are: ns (not significant), $P > 0.05$; $*P < 0.05$; $**P < 0.01$; $***P < 0.001$; $****P < 0.0001$. Source data are available online for this figure.

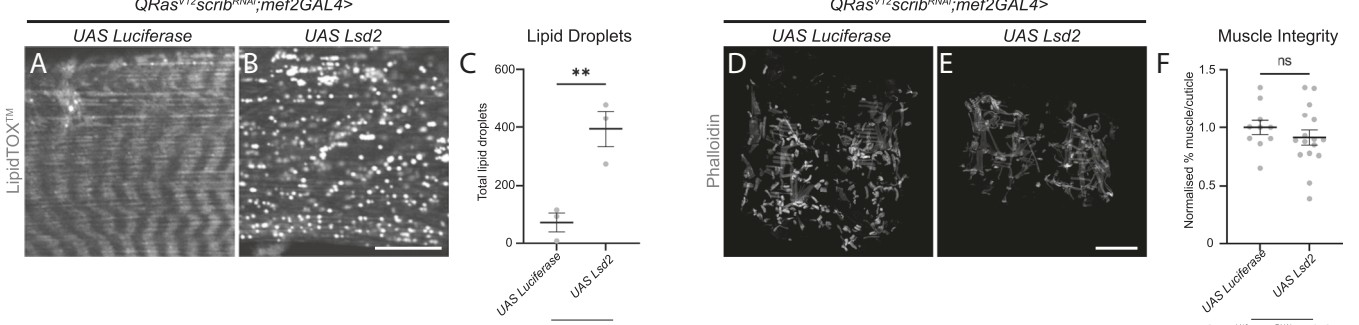

**Figure EV4.** **Increasing storage of lipids in lipid droplets does not rescue muscle integrity.**

(A, B) Lipid droplets stained with LipidTOX™ in the muscles of *QRas^V12^scrib^RNAi^;Mef2GAL4 > UAS Luciferase* and *QRas^V12^scrib^RNAi^;Mef2GAL4 > UAS Lsd2* larvae (6 AEL). (C) Quantification of the number of LDs/mm² in (A, B) performed using Student's *t* test (*n* = 3, 3). (D, E) Representative muscle fillets from *QRas^V12^scrib^RNAi^;Mef2GAL4 > UAS Luciferase* and *QRas^V12^scrib^RNAi^;Mef2GAL4 > UAS Lsd2* larvae (7 AEL), stained with Phalloidin to visualise actin. (F) Quantification of muscle integrity in N and O, performed using Student's *t* test (*n* = 10, 16). Scale bars: 20 μm for (A, B), and 500 μm for (D, E). Data information: All error bars are +/− SEM. *P* values are: ns (not significant), *P* > 0.05; **P* < 0.01. Source data are available online for this figure.

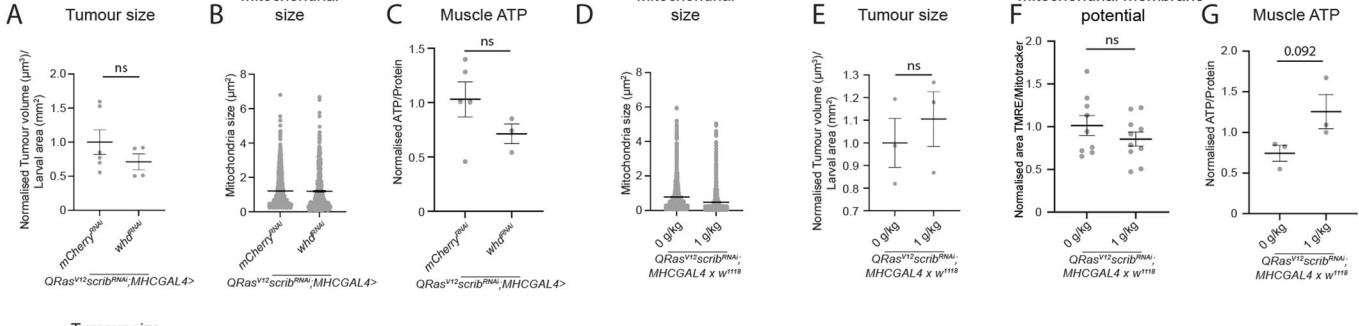

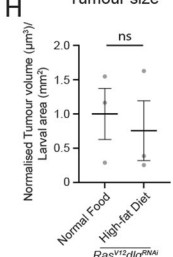

**Figure EV5. Inhibition of Whd does not significantly affect mitochondria size, ATP levels or tumour size. Mitochondria size, tumour size, mitochondria membrane potential and ATP levels under dietary manipulations.**

(A) Quantification of normalised tumour volume in $QRas^{V12}scrib^{RNAi};MhcGAL4>mCherry^{RNAi}$ and $QRas^{V12}scrib^{RNAi};MhcGAL4>whd^{RNAi}$ larvae (7 AEL), performed using Kruskal–Wallis as part of an analysis with EV2 A and EV3 F, which use the same controls ($n = 6, 4$). (B) Size distribution of individual mitochondria in the muscles of $QRas^{V12}scrib^{RNAi};MhcGAL4>mCherry^{RNAi}$ and $QRas^{V12}scrib^{RNAi};MhcGAL4>whd^{RNAi}$ larvae (6 AEL) ($n = 2037, 550$). (C) Quantification of normalised muscle ATP in $QRas^{V12}scrib^{RNAi};MhcGAL4>mCherry^{RNAi}$ and $QRas^{V12}scrib^{RNAi};MhcGAL4>whd^{RNAi}$ larvae (6 AEL), performed using one-way ANOVA as part of an analysis with EV2 C and EV3 E, which use the same controls ($n = 5, 3$). (D) Size distribution of individual mitochondria in the muscles of 7 days AEL $QRas^{V12}scrib^{RNAi};MhcGAL4>mCherry^{RNAi}$ larvae raised on a normal diet, or a diet containing 1 g/kg nicotinamide (NAM) ($n = 1447, 1454$). (E) Quantification of normalised tumour volume of 7 AEL $QRas^{V12}scrib^{RNAi};MhcGAL4 > w^{1118}$ larvae raised on a normal diet, or a diet containing 1 g/kg NAM, performed using Student's t test ($n = 3, 3$). (F) Quantification of the percentage of total mitochondria stained with MitoTracker™ Green that are shown to be active via TMRE incorporation in the muscles of 6 days AEL $QRas^{V12}scrib^{RNAi};MhcGAL4>mCherry^{RNAi}$ larvae raised on a normal diet, or a diet containing 1 g/kg NAM, performed using Student's t test ($n = 9, 10$). (G) Quantification of normalised muscle ATP in the muscles of 6 days AEL $QRas^{V12}scrib^{RNAi};MhcGAL4>mCherry^{RNAi}$ larvae raised on a normal diet, or a diet containing 1 g/kg NAM, performed using Student's t test ($n = 3, 3$). (H) Quantification of normalised tumour volume of $Ras^{V12}dlg1^{RNAi}$ larvae (8 AEL) raised on a normal diet, or a high-fat diet, performed using Student's t test ($n = 3, 3$). Data information: All error bars are $+/-$ SEM. P values are: ns (not significant). Source data are available online for this figure.

