## [Peer Review File · EMBO Reports]

Mitochondrial fusion and altered beta-oxidation drive muscle wasting in a *Drosophila* cachexia model

Callum Dark, Nashia Ali, Sofya Golenkina, Vaibhav Dhyani, Ronnie Blazev, Benjamin Parker, Kate Murphy, Gordon Lynch, Tarosi Senapati, Sean Millard, Sarah Judge, Andrew Judge, Lopamudra Giri, Sarah Russell, and Louise Cheng

Corresponding author(s): Louise Cheng (louise.cheng@petermac.org)

Review Timeline:	Transfer from Review Commons:	10th Jul 23
	Editorial Decision:	21st Jul 23
	Revision Received:	21st Nov 23
	Editorial Decision:	19th Jan 24
	Revision Received:	28th Jan 24
	Accepted:	8th Feb 24

Editor: Deniz Senyilmaz Tiebe

**Transaction Report: This manuscript was transferred to
EMBO reports following peer review at Review Commons.**

**Review
COMMONS**

Review #1

1. Evidence, reproducibility and clarity:

Evidence, reproducibility and clarity (Required)

Summary

Larvae bearing RasV12; dlGRNAi eye tumours recapitulate aspects of cachexia, such as muscle wasting. In this manuscript, the authors use their previously characterized RasV12; dlGRNAi larval model of cancer cachexia to show that tumour induced cachectic muscle wasting is associated with excessive mitochondrial fusion, resulting in the formation of enlarged dysfunctional mitochondria in wasted muscle cells. Muscle specific blockade of mitochondrial fusion prevents muscle wasting and restores mitochondrial potential in tumour bearing animals. The authors also link increased mitochondrial size to decreased insulin signaling (increased foxo) caused by the tumour induced pro-cachexia factor and insulin inhibitor Impl2. Consistently, downregulation of Impl2 from the tumour decreases foxo levels in muscle and reduces mitochondrial size. Finally, the authors show that wasting muscles in flies show decrease lipid droplets and a molecular and proteomic signature indicative of increased fatty acid oxidation. Muscle wasting, loss of lipids and mitochondrial integrity can be restored upon inhibition of Impl2 in the tumour, downregulation of the mitochondrial lipid transporter CPT1 or feeding animals with a high fat diet.

Major comments

1. All the mitochondrial phenotypes presented should be compared in the two different tumour models (Gal4/UAS and the QF/QUAS driven), which are indistinctively used throughout the study.
2. The mitochondrial phenotype of wasting muscles is only evident towards the late stages of tumorigenesis (7 day old larvae). Mitochondria of 5 day old tumour bearing animals is indistinct from the control ones. Given that 5 days is the oldest wild type larvae available, the authors need to assess the mitochondrial size and function in muscles from developmentally delayed, no-tumour bearing larvae to discard a trivial contribution of failed metamorphosis in such phenotype.
3. In all cases, the age of experimental animals must be clearly indicated in figures and/or figure legends.
4. TMRE staining presented in Figure 1 is not convincing. If available, a biochemical and/or more quantitative method to address mitochondrial function should be used.
5. Related to the point above. The extent of the mitochondrial phenotype following

genetic manipulations in the tumour or muscle is not consistently analysed. In some cases, mitochondrial size and activity is assessed but in multiple cases, only mitochondrial size is measured. Mitochondrial activity should be assessed in all cases also.

6. Are mitochondrial fusion proteins such as Marf upregulated in muscles undergoing wasting in RasV12dlg RNAi animals?
7. Is overexpression of mitochondrial fusion proteins alone sufficient to induce muscle wasting?
8. Is there a change in the expression of ATP5A in the muscles of bearing animals RasV12dlgRNAi, which has dysfunctional mitochondria compared to the control?
9. Regarding measures of insulin signaling activity in muscle (Figure 2): the data provide on FOXO staining is not very convincing. Improved staining and robust and more quantitative measure of insulin signaling activity, such as western blot analysis of pAkt should be provided. Apart from the nucleus, there is an overall increase in FOXO expression in the muscle cells of RasV12dlgRNAi compared to the control. In control animals, there is no signal of FOXO. How do you explain this?
10. The phenotype of increased fatty acid oxidation in wasting muscles is inferred as per the proteomic signature but not directly demonstrated. TCA metabolite tracing using ¹³C-Palmitate should be used to demonstrate this, which is a central point of the manuscript.
11. Does insulin signaling influence Lipid metabolism in muscle?
12. In S3 J-L, Since MHC expression is also dependent upon muscle health and integrity, it would be better to use another, and more universal, readout for protein translation/synthesis. For example, labelling the tissue with Puromycin or staining for translation initiation factors.
13. How does lipid/high fat diet restore muscle wasting? What happens to the tumours of high fat and Nicotinamide feed animals? In all cases, the impact on tumour size upon genetic manipulations of the muscle should be shown.
14. Does NAM feeding or High-fat diet restore whd transcript levels??
15. Do these feeding regimes restore insulin signaling in RasV12dlgRNAi animals?
16. The lipid phenotype in cachectic fly muscles is not consistent with that reported in humans and shown by the authors in their xenograft model. While loss of lipid droplets is observed in the fly muscle cells, there is increase in the lipid content within the mouse muscle and only extramyocellular lipid is decreased. The relevance of the extracellular lipid is unclear.
17. Related to the point above, DAPI and phalloidin should be included when showing lipid staining to understand better the cellular structures present in the field of view along with the lipid droplets.

****Minor comments****

1. The order of panels in the figures and the main text should be the same for better readability.
2. Figure S3 G-H: The image looks out of focus. Is Atg8 expression high near to the nucleus?

2. Significance:

Significance (Required)

This is an interesting study, which presents yet another mechanism involved in the regulation of tumour associated paraneoplastic syndromes, such as muscle wasting. It suggests the intriguing possibility of using a high fat diet and modulating mitochondrial metabolism as a means of alleviating cachectic muscle wasting. However, as it stands, these aspects of the study remain rather preliminary. This is particularly the case regarding the role of dietary interventions in the model and understanding of the type of metabolic reprogramming in wasting muscles, which lack direct experimental evidence. If the authors were able to further develop these aspects of the study with robust experimental work, it will make it a very valuable and impactful report.

3. How much time do you estimate the authors will need to complete the suggested revisions:

Estimated time to Complete Revisions (Required)

(Decision Recommendation)

Between 3 and 6 months

Yes

Review #2

1. Evidence, reproducibility and clarity:

Evidence, reproducibility and clarity (Required)

Chen and colleagues are using the *Drosophila* larval muscles model to investigate how a tumour can non-autonomously induce muscle mass loss, a known phenomenon called cancer cachexia. They report that tumours change muscle mitochondria morphologies, specifically their size and their chemistry. These changes correlate with increase in fat metabolism and a depletion of fat and glycogen reserves. Regarding the molecular mechanism, the authors propose that tumour cells secrete IGF binding protein that reduces the level of insulin and thus insulin signalling in muscle. They test this hypothesis by reducing FOXO activity, a negative regulator of insulin signalling, or mitochondrial fusion in muscles of tumour carrying larvae, which indeed appears to result in muscle improvements. These insights from *Drosophila* muscles suggest that tumour-caused reduced insulin signalling in muscles can be responsible for tumour induced muscle loss. A similar mechanism may apply to mammals and hence these findings are of clinical interest.

****Major comments****

1. The authors provide evidence that eye or imaginal disc tumours induce larger mitochondria in muscles. The authors try to quantify mitochondrial sizes using an automated analysis. This is a tricky task from their light microscopy images that appear to be limited in resolution. By looking at the Suppl. Figure 1, I wonder how relevant an increase of a "large" mitochondria fraction from 7 to 12 % is in the tumour larvae, considering that a significant fraction of the mitochondria are currently not counted, as they are too large to be investigated (white colours in S1F, G). Can the authors increase resolution to resolve these large clumps that likely consist of individual mitochondria to reliably segment all of them, and not only a sub fraction. It would be useful to display the size profiles of all mitochondria in various conditions and not only of a very selected subset of "large" mitochondria.

This comment applies to all figures in which mitochondria size was quantified and hence is critical for the entire manuscript.

2. Comparing MitoTracker to TMRE is a valid approach to estimate mitochondria activity/health. The images shown in 1H,I are overview images that seem to show large regional differences in the muscles of unclear origin. High resolution images of

representative regions as shown for the ATP5A stains would be more convincing as these can resolve individual mitochondria to hopefully see damaged ones next to normal ones. Would "active" mitochondria not be expected to be the ones that oxidise a lot of fatty acid break down products?

3. The authors find that co-overexpressing FOXO in muscles results in a more severe muscle degeneration phenotype in tumour bearing animals than tumour alone.

However, it seems the important control of FOXO overexpression in an otherwise wildtype animal is missing. In order to judge if the muscles really detach in these genotypes, instead of shrink and finally rupture, high resolution images of muscle attachment sites would be needed.

4. The strongly reduced lipid droplets in the tumour bearing animals is interesting. To better normalise for the reduced size of the muscles, a counter staining for muscle and a following normalisation would make the statement stronger and thus better support the conclusion.

2. Significance:

Significance (Required)

This manuscript proposes an interesting new mechanism how tumours non-autonomously induce muscle mass loss (cachexia) in a genetic *Drosophila* model. These effects can be modified by diet. Hence results are interesting for both basic and more clinically interested audience.

The weak point of the paper is the limited quantification of mitochondria sizes/morphologies, which is an important point that asks for significant improvement of either the imaging conditions or the image analysis.

3. How much time do you estimate the authors will need to complete the suggested revisions:

Estimated time to Complete Revisions (Required)

(Decision Recommendation)

Between 3 and 6 months

4. *Review Commons* values the work of reviewers and encourages them to get credit for their work. Select 'Yes' below to register your reviewing activity at Web of Science Reviewer Recognition Service (formerly Publons); note that

the content of your review will not be visible on Web of Science.

Yes

Review #3

1. Evidence, reproducibility and clarity:

Evidence, reproducibility and clarity (Required)

In this manuscript the authors study the mechanisms behind cancer cachexia, using drosophila cancer models. They find that muscle wasting in cachexia is mediated via two different mechanisms: either via insulin signalling and FOXO activation or beta oxidation via mitochondrial fusion.

It is well known that many cancers can induce a catabolic state, compatible with a decrease in insulin signalling and one of the mechanisms proposed. Additionally, the authors suggest that an imbalance between mitochondrial capacity and beta oxidation flux leads to muscle wasting.

****Major comments:****

1. Throughout the manuscript the authors use TMRE staining to evaluate mitochondrial function. To me it is not clear what function they are actually referring to. I assume they mean respiration/respiratory chain function, as this generates the proton motif force measured, but neither oxygen consumption nor aerobic ATP synthesis is ever mentioned or measured. Especially considering that the authors suggest that an increased flux through beta oxidation, which is a mitochondrial function, results in muscle wasting, the authors might want to consider measuring respiration with different substrates, using either a seahorse or Oroboros or equivalent.
2. The authors suggest that an increase in beta oxidation exceeds mitochondrial function (?), which in turn induces a change in mitochondrial morphology, further contributing to the muscle wasting. The authors may want to demonstrate that there is indeed excess beta oxidation, by measuring a toxic accumulation of different lengths of acylcarnitines. For instance, it is well known that patients with beta oxidation

defects accumulate toxic intermediates of beta oxidation that can ultimately affect mitochondrial function.

The manuscript would be much improved if oxygen consumption is measured and combined with analysis of acylcarnitines.

3. It is difficult to understand that it is even possible for beta oxidation to exceed the capacity of the OXPHOS system. In that case one would have excess of acetyl CoA and NADH, inevitably inhibiting further beta oxidation and the TCA cycle due to lack of NAD, as well as numerous regulatory mechanisms. Additionally, one would expect increased ketone body production. The authors might want to clarify how the excess redox potential, due to increased beta oxidation is utilised.

4. Unfortunately the supplementary information is in a format I can't open, thus I can't evaluate the method for identifying large mitochondria and other results in these files. This makes part of the reviewing process difficult.

****Minor:****

Line 223 "Together, this data suggests that FOXO lies upstream of beta-oxidation, and mitochondria function lies downstream of beta-oxidation".

I would suggest to rephrase. Of course beta-oxidation and the TCA takes place inside mitochondria, so what mitochondrial functions do the authors refer to?

Line 238 "Overall, this data suggests that the depletion of muscle lipid stores via beta oxidation affects mitochondrial function and is negatively correlated with muscle health in cachectic flies, mice and patients" - The mechanism is not fully clear to me as other energy sources are still available to the fly. The authors might want to expand here.

Line 93 : "To test whether this increase in mitochondrial size could lead to compromised mitochondrial function, we performed live staining with tetramethylrhodamine ethyl ester (TMRE), a compound used to measure the membrane potential of mitochondria." - I am not sure that size on its own correlates with mitochondrial function, but rather the energetic and metabolic state of the cell. Increased biogenesis is a common response to dysfunction, but often reflected in increased mass.

2. Significance:

Significance (Required)

General assessment: The strength of the study is the use of suitable in vivo modelsystems, combined with genetic manipulations to study the mechanisms behind cancer cachexia. The weak points of the study is the lack of functional assays such as quantitative measurements of oxidative phosphorylation and metabolites.

Advance: The main advance of this study is attributed to mechanistic insights behind cancer cachexia and the role of mitochondria in more conditions as opposed to the its involvement in inherited mitochondria disease.

Audience: This report should be of interest to a broad audience since it's studying a condition connected to cancer and cancer metabolism.

Reviewers field of expertise: Mitochondrial disease/dysfunction, in vivo modelling, molecular biology, bioenergetics and metabolism

3. How much time do you estimate the authors will need to complete the suggested revisions:

Estimated time to Complete Revisions (Required)

(Decision Recommendation)

Between 1 and 3 months

No

Revision Plan

Manuscript number: RC-2023-01978

Corresponding author(s): Louise Cheng

[The “revision plan” should delineate the revisions that authors intend to carry out in response to the points raised by the referees. It also provides the authors with the opportunity to explain their view of the paper and of the referee reports.]

The document is important for the editors of affiliate journals when they make a first decision on the transferred manuscript. It will also be useful to readers of the reprint and help them to obtain a balanced view of the paper.

*If you wish to submit a full revision, please use our "Full Revision" template. **It is important to use the appropriate template to clearly inform the editors of your intentions.**]*

1. General Statements [optional]

This section is optional. Insert here any general statements you wish to make about the goal of the study or about the reviews.

We would like to thank the reviewers for their insightful comments. We believe that the changes that have been suggested will add greatly to this paper, and we will endeavor to incorporate as many of these suggestions as we can.

2. Description of the planned revisions

Reviewer 1:

This is an interesting study, which presents yet another mechanism involved in the regulation of tumour associated paraneoplastic syndromes, such as muscle wasting. It suggest the intriguing possibility of using a hight fat diet and modulating mitochondrial metabolism as a means of alleviating cachectic muscle wasting. However, as it stands, these aspects of the study remains rather preliminary. This is particularly the case regarding the role of dietary interventions in the model and understanding of the type of metabolic reprogramming in wasting muscles, which lack direct experimental evidence. If the authors were able to further develop this aspects of the study with robust experimental work, it will make it a very valuable and impactful report.

1- All the mitochondrial phenotypes presented should be compared in the two different tumour models (Gal4/UAS and the QF/QUAS driven), which are indistinctively used throughout the study.

Revision Plan

We will ensure that mitochondrial size and TMRE staining are performed in the two different tumour models so that they can be compared.

2- The mitochondrial phenotype of wasting muscles is only evident towards the late stages of tumourigenesis (7 day old larvae). Mitochondria of 5 day old tumour bearing animals is indistinct from the control ones. Given that 5 days is the oldest wild type larvae available, the authors need to assess the mitochondrial size and function in muscles from developmentally delayed, no-tumour bearing larvae to discard a trivial contribution of failed metamorphosis in such phenotype.

We will examine mitochondrial size and TMRE in pmhGal4 > torso^{RNAi} animals (which undergo delayed metamorphosis) compared with control animals.

4- TMRE staining presented in Figure 1 is not convincing. If available, a biochemical and/or more quantitative method to address mitochondrial function should be used.

We will perform ATP synthesis and O² consumption assays to provide a biochemical method to accompany the TMRE assays.

5- Related to the point above. The extent of the mitochondrial phenotype following genetic manipulations in the tumour or muscle is not consistently analysed. In some cases, mitochondrial size and activity is assessed but in multiple cases, only mitochondrial size is measured. Mitochondrial activity should be assessed in all cases also.

We will assess mitochondrial activity in a time course of Ras^{V12}Dlg^{RNAi} vs w¹¹¹⁸, as well as tumor-bearing animals treated with nicotinamide, QF-QUAS Ras^{V12}scrib^{RNAi}, MHC > foxo^{RNAi}, and Ras^{V12}Dlg^{RNAi} > ImpL2^{RNAi}.

6- Are mitochondrial fusion proteins such as Marf upregulated in muscles undergoing wasting in Rasv12dlg RNAi animals?

Regulation of neither Opa1 nor Marf are altered in our proteomics study.

7- Is overexpression of mitochondrial fusion proteins alone sufficient to induce muscle wasting?

No, overexpression of Marf was not sufficient to induce muscle wasting, however overexpression of Marf caused worsened muscle wasting in tumour-bearing animals. We will include this data in our revised manuscript.

8- Is there a change in the expression of ATP5A in the muscles of bearing animals RasV12dlgRNAi, which has dysfunctional mitochondria compared to the control?

There is no change in ATP5A expression in our proteomics study.

Revision Plan

9- Regarding measures of insulin signaling activity in muscle (Figure 2): the data provide on FOXO staining is not very convincing. Improved staining and robust and more quantitative measure of insulin signaling activity, such as western blot analysis of pAkt should be provided. Apart from the nucleus, there is an overall increase in FOXO expression in the muscle cells of RasV12dlgRNAi compared to the control. In control animals, there is no signal of FOXO. How do you explain this?

We have attempted western blots of pAkt in tumour-bearing muscle previously and found that tumour metastases caused unreliable results, making immuno-staining a more reliable option. However, pAkt antibody staining also does not work well in the muscles. The control image we displayed was an extreme example, so we will choose more representative images that show more consistent FOXO staining.

12- In S3 J-L, Since MHC expression is also dependent upon muscle health and integrity, it would be better to use another, and more universal, readout for protein translation/synthesis. For example, labelling the tissue with Puromycin or staining for translation initiation factors.

We will perform O-propargyl-puromycin (OPP) staining for a w¹¹¹⁸ vs Ras^{V12}Dlg^{RNAi} time course to provide another translation readout to accompany the MHC staining.

13- How does lipid/high fat diet restore muscle wasting? What happens to the tumours of high fat and Nicotinamide feed animals? In all cases, the impact on tumour size upon genetic manipulations of the muscle should be shown.

We will measure tumour size in tumour-bearing animals on both nicotinamide and high-fat diets, as well as QF-QUAS Ras^{V12}scrib^{RNAi} MHC> foxoRNAi, marfRNAi and whdRNAi animals. Impl2RNAi in tumour-bearing animals has been shown already (Lodge et al., 2021).

14- Does NAM feeding or High-fat diet restore whd transcript levels??

We will perform qPCR to examine whd transcript levels in tumour-bearing animals on nicotinamide diets as well as high-fat diets.

15- Do these feeding regimes restore insulin signaling in RasV12dlgRNAi animals?

We have demonstrated that for Ras^{V12}dlg^{RNAi} animals fed a nicotinamide diet, FOXO levels are decreased (Fig 5D). We will do the same experiment for tumour-bearing animals fed a high fat diet.

17- Related to the point above, DAPI and phalloidin should be included when showing lipid staining to understand better the cellular structures present in the field of view along with the lipid droplets.

Revision Plan

DAPI and phalloidin staining is not compatible with lipid staining, as they require the use of PBST (detergent) which breaks down extracellular lipids. We will include more representative, raw images in which the details of the muscle can be seen.

Minor comments

1. The order of panels in the figures and the main text should be the same for better readability.

We will revisit the figures to ensure readability is improved.

2. Figure S3 G-H: The image looks out of focus. Is Atg8 expression high near to the nucleus?

Atg8a expression is highest near the nucleus, and is decreased in $Ras^{V12}dlg^{RNAi} > Impl2^{RNAi}$ animals. We will provide more representative images to make this clearer.

Reviewer 2:

This manuscript proposes an interesting new mechanism how tumours non-autonomously induce muscle mass loss (cachexia) in a genetic *Drosophila* model. These effects can be modified by diet. Hence results are interesting for both basic and more clinically interested audience.

The weak point of the paper is the limited quantification of mitochondria sizes/morphologies, which is an important point that asks for significant improvement of either the imaging conditions or the image analysis.

1. The authors provide evidence that eye or imaginal disc tumours induce larger mitochondria in muscles. The authors try to quantify mitochondrial sizes using an automated analysis. This is a tricky task from their light microscopy images that appear to be limited in resolution. By looking at the Suppl. Figure 1, I wonder how relevant an increase of a "large" mitochondria fraction from 7 to 12 % is in the tumour larvae, considering that a significant fraction of the mitochondria are currently not counted, as they are too large to be investigated (white colours in S1F, G). Can the authors increase resolution to resolve these large clumps that likely consist of individual mitochondria to reliably segment all of them, and not only a sub fraction. It would be useful to display the size profiles of all mitochondria in various conditions and not only of a very selected subset of "large" mitochondria.

This comment applies to all figures in which mitochondria size was quantified and hence is critical for the entire manuscript.

We will utilise a newly developed segmentation and centroid tracking-based analysis pipeline based in MATLAB, that may be able to separate the large clumps of mitochondria, to ensure that as many mitochondria can be quantified as possible. We will also provide size profiles of all mitochondria sizes from all conditions in which we performed mitochondria size analysis.

Revision Plan

2. Comparing MitoTracker to TMRE is a valid approach to estimate mitochondria activity/health. The images shown in 1H,I are overview images that seem to show large regional differences in the muscles of unclear origin. High resolution images of representative regions as shown for the ATP5A stains would be more convincing as these can resolve individual mitochondria to hopefully see damaged ones next to normal ones. Would "active" mitochondria not be expected to be the ones that oxidise a lot of fatty acid break down products?

We will take representative zoomed in images for 1H & I to better demonstrate mitochondria morphology.

3. The authors find that co-overexpressing FOXO in muscles results in a more severe muscle degeneration phenotype in tumour bearing animals than tumour alone. However, it seems the important control of FOXO overexpression in an otherwise wildtype animal is missing. In order to judge if the muscles really detach in these genotypes, instead of shrink and finally rupture, high resolution images of muscle attachment sites would be needed.

We will assess if $MHCGal4 > UAS\ dFOXO$ causes loss of muscle integrity. In addition, in both wildtype and tumour-bearing animals, we will overexpress FOXO in the muscles and stain for muscle attachment proteins such as tigrin to determine if the phenotype seen is caused by a mislocalisation of proteins at attachment sites.

4. The strongly reduced lipid droplets in the tumour bearing animals is interesting. To better normalise for the reduced size of the muscles, a counter staining for muscle and a following normalisation would make the statement stronger and thus better support the conclusion.

As mentioned above we will provide more representative images to help visualize muscle structures in LipidTOX experiments. In addition, we will normalize the amount of lipid droplets detected to a set area, as opposed to just measuring total lipid droplets.

Reviewer 3:

The strength of the study is the use of suitable in vivo model systems, combined with genetic manipulations to study the mechanisms behind cancer cachexia. The weak points of the study is the lack of functional assays such as quantitative measurements of oxidative phosphorylation and metabolites.

1, Throughout the manuscript the authors use TMRE staining to evaluate mitochondrial function. To me it is not clear what function they are actually referring to. I assume they mean respiration/respiratory chain function, as this generates the proton motif force measured, but neither oxygen consumption nor aerobic ATP synthesis is ever mentioned or measured. Especially considering that the authors suggest that an increased flux through beta oxidation, which is a mitochondrial function, results in muscle wasting, the authors might want to consider

Revision Plan

measuring respiration with different substrates, using either a seahorse or Oroboros or equivalent.

We do not have the necessary equipment or resources to perform Seahorse or Oroboros experiments. Therefore, we will perform O² consumption and ATP synthesis assays for Ras^{V12}dlg^{RNAi} and QF-QUAS Ras^{V12}scrib^{RNAi} vs w¹¹¹⁸, Ras^{V12}dlg^{RNAi} > Impl2^{RNAi}, QF-QUAS Ras^{V12}scrib^{RNAi} > marf^{RNAi}, whd^{RNAi}, and tumour-bearing animals fed high fat diets to provide more insights into mitochondria function.

3, It is difficult to understand that it is even possible for beta oxidation to exceed the capacity of the OXPHOS system. In that case one would have excess of acetyl CoA and NADH, inevitably inhibiting further beta oxidation and the TCA cycle due to lack of NAD, as well as numerous regulatory mechanisms. Additionally, one would expect increased ketone body production. The authors might want to clarify how the excess redox potential, due to increased beta oxidation is utilised.

We will perform acetyl-CoA and NAD/NADH assays in Ras^{V12}dlg^{RNAi} and QF-QUAS Ras^{V12}scrib^{RNAi} vs w¹¹¹⁸ to determine if beta-oxidation is occurring in excess. In addition, we will clarify in the text that we hypothesize that increased beta-oxidation is utilizing the muscle's resources to the point that there is none left to continue energy production.

Minor:

Line 223 "Together, this data suggests that FOXO lies upstream of beta-oxidation, and mitochondria function lies downstream of beta-oxidation".

I would suggest to rephrase. Of course beta-oxidation and the TCA takes place inside mitochondria, so what mitochondrial functions do the authors refer to?

As mentioned earlier, we will perform O² consumption and ATP synthesis assays to strengthen this claim. In addition, we will rephrase this sentence to avoid confusion.

Line 238 "Overall, this data suggests that the depletion of muscle lipid stores via beta oxidation affects mitochondrial function and is negatively correlated with muscle health in cachectic flies, mice and patients" - The mechanism is not fully clear to me as other energy sources are still available to the fly. The authors might want to expand here.

We will clarify that there may be other energy sources available that were not investigated in this paper.

Line 93 : "To test whether this increase in mitochondrial size could lead to compromised mitochondrial function, we performed live staining with tetramethylrhodamine ethyl ester (TMRE), a compound used to measure the membrane potential of mitochondria." - I am not sure

Revision Plan

that size on its own correlates with mitochondrial function, but rather the energetic and metabolic state of the cell. Increased biogenesis is a common response to dysfunction, but often reflected in increased mass.

We will clarify that the increase in size may be a reflection of increased metabolic need of the muscle.

3. Description of the revisions that have already been incorporated in the transferred manuscript

Please insert a point-by-point reply describing the revisions that were already carried out and included in the transferred manuscript. If no revisions have been carried out yet, please leave this section empty.

Reviewer 1:

3- In all cases, the age of experimental animals must be clearly indicated in figures and/or figure legends.

We have already put the ages of the experimental animals in the bottom of the figure legends.

11- Does insulin signaling influence Lipid metabolism in muscle?

We demonstrate in the manuscript that Foxo^{RNAi} in the muscle of tumour-bearing animals reduces whd transcript levels (Fig 4C), and Impl2^{RNAi} in the tumour restores muscle lipid droplet levels (Fig 3G-I).

4. Description of analyses that authors prefer not to carry out

Please include a point-by-point response explaining why some of the requested data or additional analyses might not be necessary or cannot be provided within the scope of a revision. This can be due to time or resource limitations or in case of disagreement about the necessity of such additional data given the scope of the study. Please leave empty if not applicable.

Reviewer 1:

10- The phenotype of increased fatty acid oxidation in wasting muscles is inferred as per the proteomic signature but not directly demonstrated. TCA metabolite tracing using 13C-Palmitate should be used to demonstrate this, which is a central point of the manuscript.

Revision Plan

The examination of ¹³C-palmitate would require metabolomic approaches, for which we do not have the necessary equipment and is beyond our timeframe. Thus, we will aim to examine changes in mitochondria metabolism through other measures mentioned above.

16- The lipid phenotype in cachectic fly muscles is not consistent with that reported in humans and shown by the authors in their xenograft model. While loss of lipid droplets is observed in the fly muscle cells, there is increase in the lipid content within the mouse muscle and only extramyo cellular lipid is decreased. The relevance of the extracellular lipid is unclear.

We hypothesize that this is due to a transport of lipids from extracellular lipid droplets to mitochondria for utilization, as has been suggested previously (Rambold et al., 2015). Examining in detail if this is the case in our models is beyond the scope of this paper.

Reviewer 3:

2, The authors suggest that an increase in beta oxidation exceeds mitochondrial function (?), which in turn induces a change in mitochondrial morphology, further contributing to the muscle wasting. The authors may want to demonstrate that there is indeed excess beta oxidation, by measuring a toxic accumulation of different lengths of acylcarnitines. For instance, it is well known that patients with beta oxidation defects accumulate toxic intermediates of beta oxidation that can ultimately affect mitochondrial function.

The manuscript would be much improved if oxygen consumption is measured and combined with analysis of acylcarnitines.

The examination of acylcarnitines would require lipidomic approaches, and is beyond our timeframe for these revisions. To try to address the need for investigations if beta-oxidation is in excess, we will perform oxygen consumption assays as mentioned and alter the manuscript to de-emphasize excess beta-oxidation.

4, Unfortunately the supplementary information is in a format I can't open, thus I can't evaluate the method for identifying large mitochondria and other results in these files. This makes part of the reviewing process difficult.

N/A

Dear Dr. Cheng,

Thank you for transferring your manuscript to EMBO Reports, which was previously reviewed at Review Commons. We concur with the referees that the proposed roles of mitochondrial fusion and fatty acid oxidation in muscle wasting are in principle interesting. However, the referees also raise significant concerns that need to be addressed to consider publication here. Having looked at all documents, we would like to invite you to submit a revised manuscript as in your revision plan. Please revise your manuscript with the understanding that the referee concerns (as in their reports) must be fully addressed and their suggestions taken on board. Please address all referee concerns in a complete point-by-point response. Acceptance of the manuscript will depend on a positive outcome of a second round of review. It is EMBO reports policy to allow a single round of major experimental revision only and acceptance or rejection of the manuscript will therefore depend on the completeness of your responses included in the next, final version of the manuscript.

We realize that it is difficult to revise to a specific deadline. In the interest of protecting the conceptual advance provided by the work, we recommend a revision within 3 months. Please discuss the revision progress ahead of this time with me if you require more time to complete the revisions, or if you have questions or comments regarding the revision (also by video chat).

1. A data availability section providing access to data deposited in public databases is missing (where applicable).
2. Your manuscript contains statistics and error bars based on $n=2$. Please use scatter plots in these cases.

You can submit the revision either as a Scientific Report or as a Research Article. For Scientific Reports, the revised manuscript can contain up to 5 main figures and 5 Expanded View figures, and it should not exceed 27000 characters. If the revision leads to a manuscript with more than 5 main figures it will be published as a Research Article. In this case the Results and Discussion section should be separate. If a Scientific Report is submitted, these sections have to be combined. This will help to shorten the manuscript text by eliminating some redundancy that is inevitable when discussing the same experiments twice. In either case, all materials and methods should be included in the main manuscript file.

3) We replaced Supplementary Information with Expanded View (EV) Figures and Tables that are collapsible/expandable online. A maximum of 5 EV Figures can be typeset. EV Figures should be cited as 'Figure EV1, Figure EV2' etc... in the text and their respective legends should be included in the main text after the legends of regular figures.

4) a .docx formatted letter INCLUDING the reviewers' reports and your detailed point-by-point responses to their comments. As part of the EMBO publication's Transparent Editorial Process, EMBO reports publishes online a Review Process File (RPF) to accompany accepted manuscripts. This File will be published in conjunction with your paper and will include the referee reports, your point-by-point response and all pertinent correspondence relating to the manuscript.

<https://www.embopress.org/page/journal/14693178/authorguide#transparentprocess>

5) a complete author checklist, which you can download from our author guidelines

<https://www.embopress.org/page/journal/14693178/authorguide>. Please insert information in the checklist that is also reflected in

the manuscript. The completed author checklist will also be part of the RPF.

6) Please note that all corresponding authors are required to supply an ORCID ID for their name upon submission of a revised manuscript (). Please find instructions on how to link your ORCID ID to your account in our manuscript tracking system in our Author guidelines

7) Before submitting your revision, primary datasets produced in this study need to be deposited in an appropriate public database (see <https://www.embopress.org/page/journal/14693178/authorguide#datadeposition>). Please remember to provide a reviewer password if the datasets are not yet public. The accession numbers and database should be listed in a formal "Data Availability" section placed after Materials & Method (see also <https://www.embopress.org/page/journal/14693178/authorguide#datadeposition>). Please note that the Data Availability Section is restricted to new primary data that are part of this study. * Note - All links should resolve to a page where the data can be accessed. *
If your study has not produced novel datasets, please mention this fact in the Data Availability Section.

Additional information on source data and instruction on how to label the files are available:
<https://www.embopress.org/page/journal/14693178/authorguide#sourcedata>

9) Our journal encourages inclusion of *data citations in the reference list* to directly cite datasets that were re-used and obtained from public databases. Data citations in the article text are distinct from normal bibliographical citations and should directly link to the database records from which the data can be accessed. In the main text, data citations are formatted as follows: "Data ref: Smith et al, 2001" or "Data ref: NCBI Sequence Read Archive PRJNA342805, 2017". In the Reference list, data citations must be labeled with "[DATASET]". A data reference must provide the database name, accession number/identifiers and a resolvable link to the landing page from which the data can be accessed at the end of the reference. Further instructions are available at <http://www.embopress.org/page/journal/14693178/authorguide#referencesformat>

12) Please also note our reference format:
<http://www.embopress.org/page/journal/14693178/authorguide#referencesformat>

I look forward to seeing a revised version of your manuscript when it is ready. Please let me know if you have questions or comments regarding the revision.

Kind regards,

Deniz Senyilmaz Tiebe

Deniz Senyilmaz Tiebe, PhD
Editor
EMBO Reports

Manuscript number: 2023-57784V1 | [RC-2023-01978]

Corresponding author(s): Louise Cheng

[The “revision plan” should delineate the revisions that authors intend to carry out in response to the points raised by the referees. It also provides the authors with the opportunity to explain their view of the paper and of the referee reports.]

The document is important for the editors of affiliate journals when they make a first decision on the transferred manuscript. It will also be useful to readers of the reprint and help them to obtain a balanced view of the paper.

*If you wish to submit a full revision, please use our "Full Revision" template. **It is important to use the appropriate template to clearly inform the editors of your intentions.**]*

1. General Statements [optional]

This section is optional. Insert here any general statements you wish to make about the goal of the study or about the reviews.

We thank the reviewer for their comments and suggestions. We have made significant improvements to our data to address the reviewer's concerns and have included the following new data:

- 1) Measured mitochondria size and mitochondria activity in the two tumour models to make them more comparable.
- 2) We have improved the quantification of mitochondria size using a new segmentation pipeline.
- 3) We have added ATP assays to examine mitochondria activity in addition to TMRE.
- 4) We have added new data to measure translation via OPP assay in addition to measuring MHC levels.
- 5) We have quantified the effects of our muscle specific genetic manipulations on tumour size, to demonstrate that the effects we see are mostly muscle-specific, and is not due to a reduction in tumour size.

2. Description of the planned revisions

Insert here a point-by-point reply that explains what revisions, additional experimentations and analyses are planned to address the points raised by the referees.

Reviewer 1:

1- All the mitochondrial phenotypes presented should be compared in the two different tumour models (Gal4/UAS and the QF/QUAS driven), which are indistinctively used throughout the study.

We have measured mitochondrial size and TMRE staining in the two different tumour models in Figures 1 and EV1.

2- The mitochondrial phenotype of wasting muscles is only evident towards the late stages of tumourigenesis (7-day old larvae). Mitochondria of 5 day old tumour bearing animals is indistinct from the control ones. Given that 5 days is the oldest wild type larvae available, the authors need to assess the mitochondrial size and function in muscles from developmentally delayed, no-tumour bearing larvae to discard a trivial contribution of failed metamorphosis in such phenotype.

We have examined mitochondrial size and TMRE in *phmGal4 > torso^{RNAi}* animals (which undergo developmental delay and do not have tumours), this data is presented in Figure EV1.

3- In all cases, the age of experimental animals must be clearly indicated in figures and/or figure legends.

We have labelled the age of the animals in the figure legends.

4- TMRE staining presented in Figure 1 is not convincing. If available, a biochemical and/or more quantitative method to address mitochondrial function should be used.

We have now performed a time course analysis for TMRE and added in zoomed in images of TMRE. In addition, we have performed ATP synthesis assays to provide a biochemical method to accompany the TMRE assays, this data is presented in Figure 1 and Figure EV1.

5- Related to the point above. The extent of the mitochondrial phenotype following genetic manipulations in the tumour or muscle is not consistently analysed. In some cases, mitochondrial size and activity is assessed but in multiple cases, only mitochondrial size is measured. Mitochondrial activity should be assessed in all cases also.

We have now assessed mitochondrial activity (TMRE) for all genetic manipulations where mitochondrial size was analysed. We have added ATP Assays to complement the TMRE assays.

6- Are mitochondrial fusion proteins such as Marf upregulated in muscles undergoing wasting in *Rasv12dlg RNAi* animals?

Regulation of neither *Opa1* nor *Marf* are altered in our proteomics study (reviewer Figure A-B).

7- Is overexpression of mitochondrial fusion proteins alone sufficient to induce muscle wasting?

No, overexpression of Marf was not sufficient to induce muscle wasting (reviewer Figure C-F), however overexpression of Marf can cause worsened muscle wasting in tumour-bearing animals (Figure EV2 D-F).

8- Is there a change in the expression of ATP5A in the muscles of bearing animals RasV12dlgRNAi, which has dysfunctional mitochondria compared to the control?

There is no change in ATP5A expression in our proteomics study (reviewer Figure G).

9- Regarding measures of insulin signaling activity in muscle (Figure 2): the data provide on FOXO staining is not very convincing. Improved staining and robust and more quantitative measure of insulin signaling activity, such as western blot analysis of pAkt should be provided. Apart from the nucleus, there is an overall increase in FOXO expression in the muscle cells of RasV12dlgRNAi compared to the control. In control animals, there is no signal of FOXO. How do you explain this?

We have attempted western blots of pAkt in tumour-bearing muscle previously and found that tumour metastases caused unreliable results, as tumours exhibit upregulated insulin signalling (Lee *et al*, 2021), therefore, any contamination from micro-metastasis in the muscle lysate confounded our results. We also tried pAkt antibody staining, however, pAkt did not work well in the muscles. Foxo staining is therefore the most reliable readout in the muscle. The control picture we had chosen previously is an extreme example. We have now included more representative example of the staining and have quantified this in Figure 2 (H-J).

11- Does insulin signaling influence Lipid metabolism in muscle?

We have demonstrated insulin signalling affects lipid metabolism. We have shown that the expression of Foxo^{RNAi} in the muscle of tumour-bearing animals reduced *whd* transcript levels (Figure 4C), and Imp12^{RNAi} in the tumour restores muscle lipid droplet levels (Figure 3 G-I).

12- In S3 J-L, Since MHC expression is also dependent upon muscle health and integrity, it would be better to use another, and more universal, readout for protein translation/synthesis. For example, labelling the tissue with Puromycin or staining for translation initiation factors.

We have performed O-propargyl-puromycin (OPP) staining for a w¹¹¹⁸ vs Ras^{V12}Dlg^{RNAi} time course to provide another translation readout to accompany the MHC staining (Figure EV3 T-V).

13- How does lipid/high fat diet restore muscle wasting? What happens to the tumours of high fat and Nicotinamide feed animals? In all cases, the impact on tumour size upon genetic manipulations of the muscle should be shown.

We have now measured tumour size under all the genetic and dietary manipulations (Figures EV 2,3,5). The effect of ImpL2RNAi on tumour size is published in (Lodge *et al*, 2021).

14- Does NAM feeding or High-fat diet restore *whd* transcript levels??

We have performed qPCR to examine *whd* transcript levels in tumour-bearing animals on nicotinamide diets as well as high-fat diets (Figure 5 H, R).

15- Do these feeding regimes restore insulin signaling in RasV12dlgRNAi animals?

We have shown the effect of NAM and high fat feeding on FOXO levels in Figure 5 D, Q.

17- Related to the point above, DAPI and phalloidin should be included when showing lipid staining to understand better the cellular structures present in the field of view along with the lipid droplets.

DAPI and phalloidin staining is not compatible with lipid staining, as they require the use of PBST (detergent) which breaks down extracellular lipids. We have included more representative images in which the details of the muscle can be better visualized (Figure 3, Figure 5 and Figure EV4).

Minor comments

1. The order of panels in the figures and the main text should be the same for better readability.

We have re-organised the panels to improve readability.

2. Figure S3 G-H: The image looks out of focus. Is Atg8 expression high near to the nucleus?

Atg8a expression is highest near the nucleus and decreased overall in Ras^{V12}dlg^{RNAi} > ImpL2^{RNAi} animals compared to control. We have provided more representative images (Figure EV3 W-Y).

Reviewer 2:

The weak point of the paper is the limited quantification of mitochondria sizes/morphologies, which is an important point that asks for significant improvement of either the imaging conditions or the image analysis.

We have made significant improvement to our image analysis using a newly developed segmentation and centroid tracking-based analysis pipeline based in MATLAB called DBSCAN.

1. The authors provide evidence that eye or imaginal disc tumours induce larger mitochondria in muscles. The authors try to quantify mitochondrial sizes using an automated analysis. This is a tricky task from their light microscopy images that appear to be limited in resolution. By looking at the Suppl. Figure 1, I wonder how relevant an increase of a "large" mitochondria fraction from 7 to 12 % is in the tumour larvae, considering that a significant fraction of the mitochondria are currently not counted, as they are too large to be investigated (white colours in S1F, G). Can the authors increase resolution to resolve these large clumps that likely consist of individual mitochondria to reliably segment all of them, and not only a sub fraction. It would be useful to display the size profiles of all mitochondria in various conditions and not only of a very selected subset of "large" mitochondria.

This comment applies to all figures in which mitochondria size was quantified and hence is critical for the entire manuscript.

We have utilized a newly developed segmentation and centroid tracking-based analysis pipeline based in MATLAB called DBSCAN, that is able to separate the large clumps of mitochondria, to ensure that as many mitochondria can be quantified as possible (Figure 1, 2, 4, 5 and EV1). We have also provided size profiles of all mitochondria sizes from conditions in which we performed mitochondria size analysis (Figure 1, 2, 4, 5, EV1, EV2, EV3 and EV5).

2. Comparing MitoTracker to TMRE is a valid approach to estimate mitochondria activity/health. The images shown in 1H,I are overview images that seem to show large regional differences in the muscles of unclear origin. High resolution images of representative regions as shown for the ATP5A stains would be more convincing as these can resolve individual mitochondria to hopefully see damaged ones next to normal ones. Would "active" mitochondria not be expected to be the ones that oxidise a lot of fatty acid break down products?

We have added representative zoomed in images for TMRE to better demonstrate mitochondria morphology (Figure EV1 D-E"). Our experimental data does not currently allow us to be able to assess the correlation between the active state of mitochondria and individual mitochondrion oxidation status.

3. The authors find that co-overexpressing FOXO in muscles results in a more severe muscle degeneration phenotype in tumour bearing animals than tumour alone. However, it seems the important control of FOXO overexpression in an otherwise wildtype animal is missing. In order to judge if the muscles really detach in these genotypes, instead of shrink and finally rupture, high resolution images of muscle attachment sites would be needed.

We have assessed if $MHCGal4 > UAS dFOXO$ causes loss of muscle integrity (Figure EV3 K-M). In tumour-bearing animals, we have overexpressed FOXO in the muscles and stained for muscle attachment proteins Tiggrin to determine if the phenotype seen is caused by a mislocalisation of proteins at attachment sites (Figure EV3 G-J). It appears that the loss of integrity is not due to the mislocalisation of Tiggrin at attachment sites.

4. The strongly reduced lipid droplets in the tumour bearing animals is interesting. To better normalise for the reduced size of the muscles, a counter staining for muscle and a following normalisation would make the statement stronger and thus better support the conclusion.

LipidTOX is incompatible with counterstains, as treatments that involved detergents disrupted lipid droplet morphology. We have provided more representative images to help visualize muscle structures in all our LipidTOX experiments (Figure 3, 5, EV4). In addition, we have normalized the amount of lipid droplets detected to a set area (Figure 3, 4, 5, EV4).

Reviewer 3:

1, Throughout the manuscript the authors use TMRE staining to evaluate mitochondrial function. To me it is not clear what function they are actually referring to. I assume they mean respiration/respiratory chain function, as this generates the proton motif force measured, but neither oxygen consumption nor aerobic ATP synthesis is ever mentioned or measured. Especially considering that the authors suggest that an increased flux through beta oxidation, which is a mitochondrial function, results in muscle wasting, the authors might want to consider measuring respiration with different substrates, using either a seahorse or Oroboros or equivalent.

The functional output of mitochondria monitored by the mitochondrial membrane potential, reflects the proton gradient generated by oxidative phosphorylation across the inner membrane. Tetramethylrhodamine ethyl ester (TMRE) is a commonly used dye to monitor mitochondria membrane potential in live cells that has been widely use in the field (Deng *et al*, 2018; Barry & Thummel).

Mitochondrial ATP production is well known to be coupled to membrane potential. In this revision, we have added new data where we performed ATP synthesis assay for Ras^{V12}dlg^{RNAi} and QF-QUAS Ras^{V12}scrib^{RNAi} vs w¹¹¹⁸, Ras^{V12}dlg^{RNAi} > Imp12^{RNAi} vs. control, QF-QUAS Ras^{V12}scrib^{RNAi} > marf^{RNAi}, whd^{RNAi}, foxo^{RNAi} vs. control and tumour-bearing animals fed high fat and nicotinamide diet.

To measure the activity of the ETC, we would need to measure O² consumption as suggested by the reviewer. This assay has been successfully performed in *Drosophila* larvae using Seahorse / Oroboros (Bülow *et al*, 2018), however, we do not have access to such equipment locally. We tried the O² consumption assay using the Extracellular Oxygen Consumption Assay (ab197243) kit, previously used in larval muscles (Song *et al*, 2017). We could not get this assay to work reproducibly in our hands, likely due to the limitations in the amount of material available.

3, It is difficult to understand that it is even possible for beta oxidation to exceed the capacity of the OXPHOS system. In that case one would have excess of acetyl CoA and NADH, inevitably inhibiting further beta oxidation and the TCA cycle due to lack of NAD, as well as numerous regulatory mechanisms. Additionally, one would expect increased ketone body production. The authors might want to clarify how the excess redox potential, due to increased beta oxidation is

utilised.

We have attempted to perform the O² consumption assay in larval muscles but have observed variability which makes the experiments hard to interpret. As we do not have the necessary equipment to perform metabolomics, we could not easily measure acetyl CoA and NADH. We no longer mention that beta oxidation exceeds the capacity of the OXPHOS system in this revision.

Minor:

Line 223 "Together, this data suggests that FOXO lies upstream of beta-oxidation, and mitochondria function lies downstream of beta-oxidation".

I would suggest to rephrase. Of course beta-oxidation and the TCA takes place inside mitochondria, so what mitochondrial functions do the authors refer to?

We have rephrased this sentence.

Line 238 "Overall, this data suggests that the depletion of muscle lipid stores via beta oxidation affects mitochondrial function and is negatively correlated with muscle health in cachectic flies, mice and patients" - The mechanism is not fully clear to me as other energy sources are still available to the fly. The authors might want to expand here.

We have rephrased this sentence.

Line 93 : "To test whether this increase in mitochondrial size could lead to compromised mitochondrial function, we performed live staining with tetramethylrhodamine ethyl ester (TMRE), a compound used to measure the membrane potential of mitochondria." - I am not sure that size on its own correlates with mitochondrial function, but rather the energetic and metabolic state of the cell. Increased biogenesis is a common response to dysfunction, but often reflected in increased mass.

We agree that the size of the mitochondrial function is not necessarily linked to mitochondrial activity, and we have rephrased this sentence.

Description of analyses that authors prefer not to carry out

Please include a point-by-point response explaining why some of the requested data or additional analyses might not be necessary or cannot be provided within the scope of a revision. This can be due to time or resource limitations or in case of disagreement about the necessity of such additional data given the scope of the study. Please leave empty if not applicable.

Reviewer 1:

10- The phenotype of increased fatty acid oxidation in wasting muscles is inferred as per the proteomic signature but not directly demonstrated. TCA metabolite tracing using ¹³C-Palmitate should be used to demonstrate this, which is a central point of the manuscript.

The examination of ¹³C-palmitate would require metabolomic approaches, for which we do not have the necessary equipment and is beyond our timeframe.

16- The lipid phenotype in cachectic fly muscles is not consistent with that reported in humans and shown by the authors in their xenograft model. While loss of lipid droplets is observed in the fly muscle cells, there is increase in the lipid content within the mouse muscle and only extramyocellular lipid is decreased. The relevance of the extracellular lipid is unclear.

We hypothesize that this is due to a transport of lipids from extracellular lipid droplets to mitochondria for utilization, as has been suggested previously (Rambold *et al*, 2015). Examining in detail if this is the case in our models is beyond the scope of this paper.

Reviewer 3:

2, The authors suggest that an increase in beta oxidation exceeds mitochondrial function (?), which in turn induces a change in mitochondrial morphology, further contributing to the muscle wasting. The authors may want to demonstrate that there is indeed excess beta oxidation, by measuring a toxic accumulation of different lengths of acylcarnitines. For instance, it is well known that patients with beta oxidation defects accumulate toxic intermediates of beta oxidation that can ultimately affect mitochondrial function.

The manuscript would be much improved if oxygen consumption is measured and combined with analysis of acylcarnitines.

The examination of acylcarnitines would require lipidomic/metabolomics approaches, which we do not have the necessary equipment to conduct these experiments. We have withdrawn the text on "beta oxidation exceeds mitochondria function". We did attempt the O² consumption assay in larval muscles but have observed variability which makes the experiments hard to interpret. We no longer mention that beta oxidation exceeds the mitochondrial function in this revision.

4, Unfortunately the supplementary information is in a format I can't open, thus I can't evaluate the method for identifying large mitochondria and other results in these files. This makes part of the reviewing process difficult.

N/A

Reviewer Figure

A) Quantification of Log transformed Opa1 protein levels, from proteomics performed in Figure 4 B, performed using Mann-Whitney U. There are no significant differences in Opa1 levels between control (5 AEL) and *Ras^{V12}Dig^{RNAi}* animals (7 AEL, n = 5, 4).

B) Quantification of Log transformed Marf protein levels, from proteomics performed in Figure 4 B, performed using Mann-Whitney U. There are no significant differences in Marf levels between control (5 AEL) and *Ras^{V12}Dig^{RNAi}* animals (7 AEL, n = 5, 4).

C, D) Muscle fillets from 5 AEL *UAS Mito-GFP;MHC GAL4>UAS Luciferase* and *UAS Mito-GFP;MHC GAL4>UAS marf* larvae, stained with Phalloidin to visualise actin. *UAS marf* muscle shows no obvious signs of degradation.

E, F) Images of mitochondria tagged with GFP in the muscles of 5 AEL *UAS Mito-GFP;MHC GAL4>UAS Luciferase* and *UAS Mito-GFP;MHC GAL4>UAS marf* larvae. *UAS marf* muscle appears to have larger, circular mitochondria than control.

G) Quantification of Log transformed ATP5A protein levels, from proteomics performed in Figure 4 B, performed using Mann-Whitney U. There are no significant differences in ATP5A levels between control (5 AEL) and *Ras^{V12}Dig^{RNAi}* animals (7 AEL, n = 5, 4).

Scale Bars: 10 μ m for (E and F), and 500 μ m for (C and D).

All error bars are +/- SEM. P values are: ns (not significant).

- Barry WE & Thummel CS The Drosophila HNF4 nuclear receptor promotes glucose-stimulated insulin secretion and mitochondrial function in adults. *eLife* 5: e11183
- Bülow MH, Wingen C, Senyilmaz D, Gosejacob D, Sociale M, Bauer R, Schulze H, Sandhoff K, Teleman AA, Hoch M, *et al* (2018) Unbalanced lipolysis results in lipotoxicity and mitochondrial damage in peroxisome-deficient Pex19 mutants. *Mol Biol Cell* 29: 396–407
- Deng H, Takashima S, Paul M, Guo M & Hartenstein V (2018) Mitochondrial dynamics regulates Drosophila intestinal stem cell differentiation. *Cell Death Discov* 4: 1–13
- Lee J, Ng KG-L, Dombek KM, Eom DS & Kwon YV (2021) Tumors overcome the action of the wasting factor ImpL2 by locally elevating Wnt/Wingless. *Proc Natl Acad Sci* 118: e2020120118
- Lodge W, Zavortink M, Golenkina S, Frolidi F, Dark C, Cheung S, Parker BL, Blazev R, Bakopoulos D, Christie EL, *et al* (2021) Tumor-derived MMPs regulate cachexia in a Drosophila cancer model. *Dev Cell* 56: 2664-2680.e6
- Rambold AS, Cohen S & Lippincott-Schwartz J (2015) Fatty Acid Trafficking in Starved Cells: Regulation by Lipid Droplet Lipolysis, Autophagy, and Mitochondrial Fusion Dynamics. *Dev Cell* 32: 678–692
- Song W, Owusu-Ansah E, Hu Y, Cheng D, Ni X, Zirin J & Perrimon N (2017) Activin signaling mediates muscle-to-adipose communication in a mitochondria dysfunction-associated obesity model. *Proc Natl Acad Sci U S A* 114: 8596–8601

Dear Louise,

Thank you for submitting your revised manuscript. It has now been seen by all of the original referees.

My apologies for the delay in getting back to you - it took longer than anticipated to receive the referee reports given this busy time of the year.

As you can see, the referees find that the study is significantly improved during revision and recommend publication. However, I need you to address the points below before I can accept the manuscript.

- Please address the remaining concerns of referees #1 and #2. Please let me know if you would like to discuss any of the points further.
- Please rename the 'Declaration of Interests' as 'Disclosure Statement and Competing Interests'.
- Please rename the 'Methods' as 'Materials and Methods'.
- We note a mismatch in one of the author names between our manuscript tracking system and the manuscript (Sofia Golenkina vs. Sofya Golenkina).
- Please remove the 'Author contribution' section from the manuscript.
- As per our format requirements, in the reference list, citations should be listed in alphabetical order and then chronologically, with the authors' surnames and initials inverted; where there are more than 10 authors on a paper, 10 will be listed, followed by 'et al.'. Please see <https://www.embopress.org/page/journal/14693178/authorguide#referencesformat> We note that some references have DOIs, which are only needed for preprints and datasets that have not been published yet.
- We note that Bakopoulos et al., 2023, is currently cited as preprint, which was in the meantime published. Please update the citation accordingly. Also, please contact bioRxiv to link the preprint to the publication, which is the version of record.
- We note the presence of the phrase 'Data not shown' on page 7, which is not allowed as per EMBO Press policies (<https://www.embopress.org/page/journal/14693178/authorguide#unpublisheddata>). Please either show the data or remove the statement.
- Please fill out and include an author checklist as listed in our online guidelines (<https://www.embopress.org/page/journal/14693178/authorguide>)
- We note that Figure 5T is currently not called out in the text.
- The manuscript sections should be in the following order: Title page - Abstract & Keywords - Introduction - Results - Discussion - Materials & Methods - Data Availability - Acknowledgments - Disclosure Statement & Competing Interests - References - Figure Legends - Tables with legends - Expanded View Figure Legends.
- Please resubmit the EV Macros 1 and 2 as Computer Code EV1 and Computer Code EV2, respectively. Please supply individual ZIP files containing the data file (macro) and a separate plain text README file with item title and description. Please submit these using the file type Expanded View File in our manuscript submission system and update the text callouts accordingly.
- The source data of the main figure should be uploaded as one zip file per figure. Source data for EV figures can be grouped into one folder.
- Please add the title of 'Expanded View figure legends'.
- Our production/data editors have asked you to clarify several points in the figure legends:
 - Please note that a separate 'Data Information' section is required in the legends of all the figures. (please see <https://www.embopress.org/page/journal/14693178/authorguide#figureformat> for examples)
 - Please indicate the statistical test used for data analysis in the legends of figures 4a
- Papers published in EMBO Reports include a 'synopsis' and 'bullet points' to further enhance discoverability. Both are displayed on the html version of the paper and are freely accessible to all readers. The synopsis includes a short standfirst summarizing the study in 1 or 2 sentences (max 35 words) that summarize the paper and are provided by the authors and streamlined by the handling editor. I would therefore ask you to include your synopsis blurb and 3-5 bullet points listing the key experimental findings.
- In addition, please provide an image for the synopsis. This image should provide a rapid overview of the question addressed in the study but still needs to be kept fairly modest since the image size cannot exceed 550 (width) x 300-600 (height) pixels.

Thank you again for giving us to consider your manuscript for EMBO Reports, I look forward to your minor revision.

Kind regards,

Deniz

--

Deniz Senyilmaz Tiebe, PhD
Editor
EMBO Reports

Referee #1:

I would like to thank the authors for the revised manuscript. I have some minor comments that nevertheless are important.

1. Do not use the expression "mitochondrial activity" when you have not measured that, but rather the mitochondrial membrane potential. It is not the same, please rephrase. For example see row 107-111. Please also change all figures where TMRE/mitotracker results are shown and the title "mitochondrial activity" used to instead be titled "mitochondrial membrane potential" or similar.
2. Mitochondria do not produce ATP they perform oxidative phosphorylation via OXPHOS, making them the main site for cellular energy conversion. Energy cannot be produced only converted (see row 112-113 for example).

Referee #2:

This revised manuscript is improved. I want to thank the authors for taking some of my suggestions into account. I still feel that some important issues remain to be addressed.

1. The authors have now significantly improved their mitochondria size quantification methods. I am now convinced by the data presented that the tumors do increase the size of the muscle mitochondria. I still feel the presentation of the size differences could be strongly improved, currently only 3 bins are presented. Why not showing many more and overlay the different conditions in different colours? Such a shift of the distributions would be more obvious. Which 2 conditions were compared to calculate the reported p-value with Chi-square in Figure 4? Only 4 larvae per condition of this key experiment is a low number that could be improved.
2. In my initial review, I have asked for high resolution images of TMRE/Mitotracker, to better assess how these stainings can be compared and hence how ratio changes can be interpreted. The current images are improved but are still overview images, I cannot see the larger mitochondria, in fact in most images it is even hard to see the muscles. Hence, the current data do not support the strong conclusion of the authors. I would like to see similarly high-resolution images as shown now for Figure 1E, F, ideally using the segmentation tool to remove all the background and only quantify mitochondria signal. Same applies for Fig. 1U. If this is not possible, conclusions need to be toned down.
3. An important point not mentioned in my initial review: the authors suggest in Fig 1N-Q that blocking mitochondria fusion rescues muscle integrity. This is likely the case. However, I am not convinced by the presented data. How is the value of 1 in the tumour model calculated? Does this take into account the percentage of 'detached' muscles? I could not find information in the methods. This detachment seems to largely occur at the edges of the fillets, thus is likely an artifact of the preparation. It would be more convincing to use intact larvae and quantify 'detachment' by imaging a muscle GFP marker or UAS-GFP. Same applies to Fig. 1V-X. Together with the following point, it might be more informative to quantify muscle volume, instead of 'detachment', which in my opinion does not occur. Muscle volume is also more relevant to cachexia.
4. The authors have now done Tigrin staining and conclude that Tigrin is normal in the tumour but muscles 'detach' in the tumour. How can both be possible together? Either muscles detach or Tigrin at the attachments is normal. It seems to me that these muscles rather degenerate and undergo atrophy, but do not undergo force-induced detachment, at least not in the classical sense as described in many of the integrin mutants, which as a consequence of course affect the attachments. The only high-resolution muscle morphology images shown in the entire paper are the ones in which Tigrin has been stained for. And in these it seems that fragments muscles REMAIN attached. Thus, fibers are rather thinning and degenerating.

Minor:

1. Please follow Flybase gene names: Mhc, Mef2, dlg1 (not MHC or mef2, dlg).
2. Labelling the figures with numbers in the PDF would save reviewers' time.

Referee #3:

I have looked at the revised manuscript and believe that this revised version as per my original comments has greatly improved the content and quality of the work. The manuscript is, in my opinion, now suitable for publication.

- Please rename the 'Declaration of Interests' as 'Disclosure Statement and Competing Interests'.
done
- Please rename the 'Methods' as 'Materials and Methods'.
done
- We note a mismatch in one of the author names between our manuscript tracking system and the manuscript (Sofia Golenkina vs. Sofya Golenkina).
Done
- Please remove the 'Author contribution' section from the manuscript.
Done
- As per our format requirements, in the reference list, citations should be listed in alphabetical order and then chronologically, with the authors' surnames and initials inverted; where there are more than 10 authors on a paper, 10 will be listed, followed by 'et al.'. Please see <https://www.embopress.org/page/journal/14693178/authorguide#referencesform> at We note that some references have DOIs, which are only needed for preprints and datasets that have not been published yet.
Done
- We note that Bakopoulos et al., 2023, is currently cited as preprint, which was in the meantime published. Please update the citation accordingly. Also, please contact bioRxiv to link the preprint to the publication, which is the version of record.
Done
- We note the presence of the phrase 'Data not shown' on page 7, which is not allowed as per EMBO Press policies (<https://www.embopress.org/page/journal/14693178/authorguide#unpublisheddata>). Please either show the data or remove the statement.
Done, we have removed the statement.
- Please fill out and include an author checklist as listed in our online guidelines (<https://www.embopress.org/page/journal/14693178/authorguide>)
- We note that Figure 5T is currently not called out in the text.
There is no Figure 5T.
- The manuscript sections should be in the following order: Title page - Abstract & Keywords - Introduction - Results - Discussion - Materials & Methods - Data Availability - Acknowledgments - Disclosure Statement & Competing Interests - References - Figure Legends - Tables with legends - Expanded View Figure Legends.
Done
- Please resubmit the EV Macros 1 and 2 as Computer Code EV1 and Computer Code EV2, respectively. Please supply individual ZIP files containing the data file (macro) and a separate plain text README file with item title and description. Please submit these using the file type Expanded View File in our manuscript submission system and update the text callouts accordingly.
Done
- The source data of the main figure should be uploaded as one zip file per figure (2mb) Source data for EV figures can be grouped into one folder.
Done
- Please add the title of 'Expanded View figure legends'.

Done.

- Our production/data editors have asked you to clarify several points in the figure legends:
 - Please note that a separate 'Data Information' section is required in the legends of all the figures. (please see <https://www.embopress.org/page/journal/14693178/authorguide#figureformat> for examples)

Done.

- Please indicate the statistical test used for data analysis in the legends of figures 4a

Done

- Papers published in EMBO Reports include a 'synopsis' and 'bullet points' to further enhance discoverability. Both are displayed on the html version of the paper and are freely accessible to all readers. The synopsis includes a short standfirst summarizing the study in 1 or 2 sentences (max 35 words) that summarize the paper and are provided by the authors and streamlined by the handling editor. I would therefore ask you to include your synopsis blurb and 3-5 bullet points listing the key experimental findings.

Done

- In addition, please provide an image for the synopsis. This image should provide a rapid overview of the question addressed in the study but still needs to be kept fairly modest since the image size cannot exceed 550 (width) x 300-600 (height) pixels.

Done

Thank you again for giving us to consider your manuscript for EMBO Reports, I look forward to your minor revision.

Kind regards,

Deniz

--

Deniz Senyilmaz Tiebe, PhD
Editor
EMBO Reports

Referee #1:

I would like to thank the authors for the revised manuscript. I have some minor comments that nevertheless are important.

1. Do not use the expression "mitochondrial activity" when you have not measured that, but rather the mitochondrial membrane potential. In is not the same, please rephrase. For example see row 107-111. Please also change all figures where TMRE/mitotracker results are shown and the titel "mitochondrial activity" used to instead be titled "mitochondrial membrane potential" or similar.

We agree with the suggestion and have changed the text.

2. Mitochondria do not produce ATP they perform oxidative phosphorylation via OXPHOS, making them the main site for cellular energy conversion. Energy cannot be produced only converted (see row 112-113 for example).

We agree with reviewer's suggestion and have changed the text.

Referee #2:

This revised manuscript is improved. I want to thank the authors for taking some of my suggestions into account. I still feel that some important issues remain to be addressed.

1. The authors have now significantly improved their mitochondria size quantification methods. I am now convinced by the data presented that the tumors do increase the size of the muscle mitochondria. I still feel the presentation of the size differences could be strongly improved, currently only 3 bins are presented. Why not showing many more and overlay the different conditions in different colours? Such, a shift of the distributions would be more obvious.

We have presented the raw data of all the muscle mitochondria in EV files, which demonstrates the distribution of the size differences.

We beg to differ with the reviewer that the presentation all the value would add clarity to the data, so we are keeping the data presentation as it is.

Which 2 conditions were compared to calculate the reported p-value with Chi-square in Figure 4?

In the methods section we stated: "The percentage of mitochondria in each of these categories was averaged across replicates, and the distribution of mitochondrial size across categories was compared via Chi-square test." We are comparing the shift changes in mitochondria size amongst the three categories.

Only 4 larvae per condition of this key experiment is a low number that could be improved.

We have performed n=5 for this experiment. We think the reviewer is questioning if we used greater number of larvae, whether we would then detect a significant difference. We would like to point out in other experimental settings, we have detected significant differences using n=5, so we believe a greater n number here would not make a difference to our analysis.

2. In my initial review, I have asked for high resolution images of TMRE/Mitotracker, to better assess how these stainings can be compared and hence how ratio changes can be interpreted. The current images are improved but are still overview images, I cannot see the larger mitochondria, in fact in most images it is even hard to see the muscles. Hence, the current data do not support the strong conclusion of the authors. I would like to see similarly high-resolution images as shown now for Figure 1E, F, ideally using the segmentation tool to remove all the background and only quantify mitochondria signal. Same applies for Fig. 1U. If this is not possible, conclusions need to be toned down.

Figure 1E, F are ATF5A staining which stains for mitochondria. We believe we have

supplied the high resolution pictures of mitochondria in Figures 1R,S, Figure 2 A, B etc, as requested by the reviewer.

We think the reviewer is asking how we can calculate TMRE if we can not see individual mitochondria. We would like to clarify that when we measure mitochondria activity (membrane potential), we are measuring the area of TMRE as a percentage of the area of mitotracker. We are not looking at the intensity of individual mitochondria, because that varies greatly across the muscle fillet. Instead, we are calculating mitochondria membrane potential across the whole fillet. We have clarified this by changing the Y axis of all graphs that quantify mitochondria membrane potential to “Normalised area TMRE /Mitotracker (%)”.

3. A important point not mentioned in my initial review: the authors suggest in Fig 1N-Q that blocking mitochondria fusion rescues muscle integrity. This is likely the case. However, I am not convinced by the presented data. How is the value of 1 in the tumour model calculated?

In the methods we stated: percentage muscle/cuticle was determined using FIJI as previously described(Lodge *et al*, 2021; Dark *et al*, 2022). In brief, dissected muscle fillets stained with Phalloidin to mark actin were analysed using a FIJI macro(Dark *et al*, 2022). A ROI was drawn around the cuticle of the muscle fillet, and the image was converted to a binary mask using the “Auto Threshold” tool. The total area of fluorescence detected within the ROI was divided by the total ROI area, which we calculated as % muscle attachment.

To obtain the value of 1, we normalised all the data points in control to the average of the control.

Does this take into account the percentage of 'detached' muscles? I could not find information in the methods.

We refer the reviewer to Dark 2022 (Star Methods) and Lodge 2021 (Dev Cell) where the below figure from Dark 2022 illustrates how the macro calculates muscle degradation.

Download : Download high-res image (617KB)

Download : Download full-size image

Figure 4. Examples of outputs from the Fiji muscle fillet macro

(A and B) Representations of different ROI boundaries.

(C) Example of an intact fillet. Scale bar= 500µm.

(D) Example of a detached fillet. Scale bar= 500µm.

(E) Quantification of fillets from C & D. Error bars=± SEM, * = $p < 0.05$.

It would be more convincing to use intact larvae and quantify 'detachment' by imaging a muscle GFP marker or UAS-GFP. Same applies to Fig. 1V-X. Together with the following point, it might more informative to quantify muscle volume, instead of 'detachment', which in my opinion does not occur. Muscle volume is also more relevant to cachexia.

We refer the reviewer to Figure 2, Figure S1 of Lodge et al., 2021 (Dev Cell), where we have quantified muscle degradation in intact larvae by imaging a muscle GFP marker (Zasp-GFP). We showed that muscle integrity measured as % muscle/cuticle is reflective of the amount of muscle present. We agree the term muscle detachment is not accurate, and we have changed the wording "detachment" to "degradation" throughout the manuscript.

This detachment seems to largely occur at the edges of the fillets, thus is likely an artifact of the preparation.

We disagree. Degradation is not occurring at the edges. Our methods take into account of all muscle, attached or degraded.

4. The authors have now done Tiggrin staining and conclude that Tiggrin is normal in the tumour but muscles 'detach' in the tumour.

We have changed the wording from 'detachment' to 'degradation' throughout the manuscript.

How can both be possible together? Either muscles detach or Tiggrin at the attachments is normal. It seems to me that these muscles rather degenerate and undergo atrophy, but do not undergo force-induced detachment, at least not in the classical sense as described in many of the integrin mutants, which as a consequence of course affect the attachments.

We agree it is likely atrophy. In our previous publications, Lodge et al., 2021 (Dev Cell) and Bakopoulos et al., 2023 (EMBO reports) we have characterised at length what causes the muscle degradation phenotype. Our data suggests that muscle degradation encompasses both atrophy as well as ECM mediated detachment at the muscle/tendon junction. As FOXO impinges on insulin signalling, it likely it is atrophy. We have made this point more explicit in the text.

The only high-resolution muscle morphology images shown in the entire paper are the ones in which Tigrin has been stained for. And in these it seems that fragments muscles REMAIN attached. Thus, fibers are rather thinning and degenerating.

Minor:

1. Please follow Flybase gene names: Mhc, Mef2, dlg1 (not MHC or mef2, dlg).

corrected

2. Labelling the figures with numbers in the PDF would save reviewers' time.

Corrected

Dear Louise,

Thank you for submitting your revised manuscript. I have now looked at everything and all is fine. Therefore, I am very pleased to accept your manuscript for publication in EMBO Reports.

Congratulations on a nice work!

Kind regards,

Deniz

--

Deniz Senyilmaz Tiebe, PhD

Editor

EMBO Reports
